# Use-Case-Grounded Simulations for Explanation Evaluation

**Valerie Chen**[*] **Nari Johnson  Nicholay Topin**[†] **Gregory Plumb**[†] **Ameet Talwalkar**
Carnegie Mellon University

## Abstract

A growing body of research runs human subject evaluations to study whether providing users with explanations of machine learning models can help them with practical real-world use cases. However, running user studies is challenging and costly, and consequently each study typically only evaluates a limited number of different settings, e.g., studies often only evaluate a few arbitrarily selected explanation methods. To address these challenges and aid user study design, we introduce Use-Case-Grounded **Sim**ulated **Eval**uations (SimEvals). SimEvals involve training algorithmic agents that take as input the information content (such as model explanations) that would be presented to each participant in a human subject study, to predict answers to the use case of interest. The algorithmic agent's test set accuracy provides a measure of the predictiveness of the information content for the downstream use case. We run a comprehensive evaluation on three real-world use cases (forward simulation, model debugging, and counterfactual reasoning) to demonstrate that SimEvals *can* effectively identify which explanation methods will help humans for each use case. These results provide evidence that SimEvals can be used to efficiently screen an important set of user study design decisions, e.g. selecting which explanations should be presented to the user, before running a potentially costly user study.

## 1 Introduction

The field of interpretable machine learning has proposed a large and diverse number of techniques to explain model behavior (e.g., [37, 28, 38, 34, 29]). A growing body of work studies which of these techniques can help humans with practical real-world use cases [2, 20, 16, 18, 33, 35, 8, 16]. Because it is difficult to anticipate exactly which explanations may help humans with a particular use case [9, 10], the gold standard to evaluate an explanation's usefulness to humans is to run a *human subject study* [11]. These studies ask users a series of questions about the use case and provide the users with relevant information (such as model explanations) to answer each question.

Selecting the relevant information, which we call *information content*, to provide each user in the human subject study comprises an important set of design decisions [18, 5]. The researcher needs to consider the choice of explanation methods, hyperparameters that are used to calculate the model explanations, and whether additional pieces of information should be provided to the user (such as the model's prediction on the data-point being explained).[1] While a researcher may wish to run several user studies to evaluate many different types of information content, in practice this is often infeasible: user studies are resource intensive and time-consuming, requiring the recruitment and

---

[*]Correspondence to `valeriechen@cmu.edu`.

[†]Equal Contribution

[1]We note that there are several additional user study design decisions beyond *what information* is shown to the user: for example, how the information is presented or visualized, the user study interface design, etc. We consider these additional design decisions to be out-of-scope for our study for reasons discussed in Appendix J.

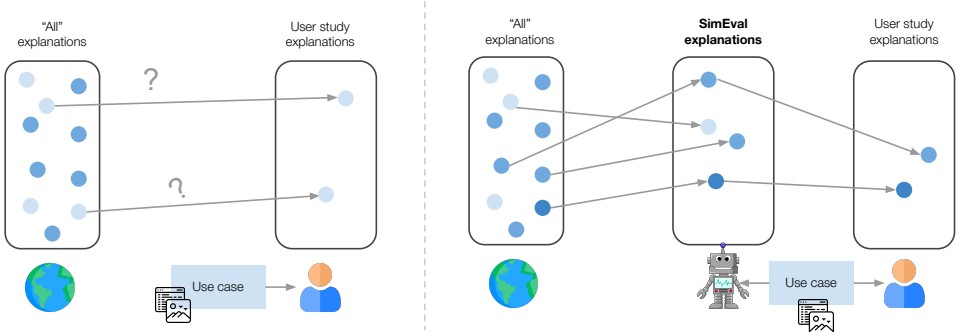

Figure 1: An overview of how `SimEvals` can help a researcher with an important user study design decision: selecting which explanation methods to evaluate given their specific *use case*. (Left) Prior to our work, existing user studies often only evaluate a small number of explanation methods due to resource constraints. When selecting candidate explanations to evaluate, researchers often simply choose the most popular or well known explanation methods with little justification about *why* each explanation may be helpful for the downstream use case. (Right) We propose using `SimEvals`, which are use-case-grounded, algorithmic evaluations, to efficiently screen explanations *before* running a user study. In this example, the researcher runs a `SimEval` on each of the four candidate explanation methods and then uses the results of the `SimEvals` to select two promising explanation methods where the algorithmic agent has high accuracy for their human subject study.

compensation of real users. Due to these constraints, most studies are far from comprehensive in terms of the number of design decisions evaluated. For example, most user studies typically only evaluate a few explanation methods, with default hyperparameter settings, that are often arbitrarily selected based on their perceived popularity (Figure 1, Left).

For example, forward simulation is a canonical use case where users are asked "Can you predict what the model's prediction is for this individual?" [35, 8, 16]. To answer this question, the user is provided with information content, which may include data-points (e.g., the individual's features) and model explanations for those data-points (e.g., a SHAP [29] importance score for each feature). The researcher can then measure which set of information content (and consequently which *explanation method*) enables users to achieve the best performance (e.g. the highest accuracy when answering the use case questions). Beyond forward simulation, other user studies have investigated use cases ranging from model debugging to counterfactual reasoning [2, 20, 16, 18, 33].

In this work, we introduce Use-Case-Grounded **Sim**ulated **Eval**uations (`SimEvals`): a general framework for conducting use-case-grounded, algorithmic evaluations of the information content present in a user study (Figure 1, Right). Each `SimEval` involves training and evaluating an algorithmic agent, which is an ML model. The agent learns to predict the ground truth label for the use case given the same information that would be presented to a user in a human subject study. The agent's test set accuracy is a measure of whether the information content provided (which critically includes the model explanation) is predictive for the use case.

By comparing the agent's test set accuracy for different types of information content, the researcher can efficiently identify promising and eliminate unhelpful information content types (e.g. explanation methods as shown in Figure 1) to present to real humans in their final human subject study. Since `SimEvals` are intended to inform, *not* replace, human subject evaluations, we note that our algorithmic agent does not incorporate additional factors such as the effects of cognitive biases [26, 35], that may affect human decision-making (see Appendix J). In our experiments, we show that using `SimEvals` to measure predictive information *can* indeed identify which information content can help a human with a downstream use case.

In this work, we focus primarily on the design choice of selecting which explanation methods to evaluate. We conduct an evaluation on a diverse set of use cases and explanation methods: (1) We compare `SimEvals` with findings from prior user studies, and (2) we run our own user study motivated by use cases studied in prior work. In both cases, we find that `SimEval` effectively distinguishes between explanations that are ultimately helpful versus unhelpful to human subjects in the user study.

Our primary contributions in this work are as follows:

1. **Introducing `SimEvals` Framework.** We are the first to propose an automated use-case-grounded evaluation framework to measure the predictive information contained in the information content provided to a user in a human subject study. We instantiate our framework on three diverse use cases to illustrate important differences in how `SimEvals` can be instantiated, showcasing how the `SimEval` framework covers a wide variety of real-world use cases and explanation methods.

2. **Identifying Promising Explanations for Humans.** We demonstrate that `SimEvals` can distinguish between which explanations are promising versus unhelpful for humans on each of the three use cases we consider. Specifically, we find that when there is a significant gap between `SimEval` performance for two explanations, we observe a similar significant gap in human performance when using the same explanations.

## 2 Related Work

**Types of User Studies on Explanations:** Many prior works run user studies to evaluate IML methods, covering a diverse set of goals [30]. Specifically, IML user studies can differ in the independent variable that is changed (e.g., the explanation method used [20, 16] or the visualization of the explanation [24]) and the dependent variable(s) being measured (e.g., the user's performance on a use case [20, 16, 26, 28, 27, 14], time taken to complete the use case [26], or trust in the ML model [7]). We focus on the set of user studies that measure how different *explanation methods* (independent variable) affect how *accurately* users can complete a use case (dependent variable).

We group existing user studies on explanations that measure the accuracy of users on a use case into two categories. The first category studies the utility of explanations in use cases where the *predictiveness* of the explanation method is already known to researchers. For example, [26, 28, 27] ask users to answer simple comprehension questions about a decision set explanation, where these explanations contain the information necessary to complete the use case with perfect accuracy. Thus, these studies measure humans' ability to comprehend and interpret the explanations without error. In contrast, the second category includes studies where the predictiveness of the explanation method is *not* known to researchers in advance of their study [24, 16, 2]. Our framework was developed to aid researchers in this second setting to quickly validate whether or not an explanation method is predictive for their use case.

**Algorithmic frameworks for evaluating explanations:** One line of evaluation frameworks focuses on automating the ability to measure desirable properties relating to the faithfulness of the explanation to the model [22, 1, 44], which does not account for any downstream use cases for the explanation. In contrast, our algorithmic evaluation framework allows researchers to customize and perform a *use case grounded* evaluation of explanation methods.

We were able to identify one-off examples that use algorithmic agents to perform use case grounded evaluations in prior work [33, 45, 23, 36]. All of these examples propose evaluations designed around one specific use case. Our framework is more general, and can be applied to a wide variety of use cases for explanations. Our work is the first to propose a learning-based agent to identify which explanation methods may be predictive for a downstream use case. We provide a more in-depth discussion of how these examples can be contextualized in our general framework in Appendix A.

## 3 General Framework

We present a general framework to train `SimEvals`[2], which measure the predictiveness of information content choices for a downstream use case. Recall that when designing a user study, the researcher must identify candidate choices of information content (e.g. which explanation methods, hyperparameters, or other additional baseline information) to evaluate. Our framework trains a `SimEval` agent for *each* choice of information content that the researcher intends to evaluate for a downstream use case. The researcher can then interpret the test set accuracy of each `SimEval` agent as a measure of the predictiveness of that information content. Before describing how to train each agent, we overview three use-case-specific components *shared* across all agents/types of information content that the researcher must instantiate:

---

[2]Tutorial on how to run your own `SimEvals`: `https://github.com/valeriechen/simeval_tutorial`

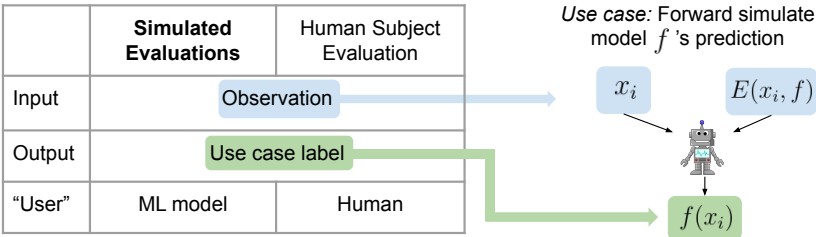

| | Simulated Evaluations | Human Subject Evaluation |
|---|---|---|
| Input | Observation | |
| Output | Use case label | |
| "User" | ML model | Human |

Figure 2: (Left) `SimEval` trains an agent (e.g., ML model) on the same information (observation and use case label) that a user would see in a human subject evaluation. (Right) For the forward simulation use case, where the goal is to simulate the ML model's prediction for a given data-point, we demonstrate how to instantiate the observation (which contains the data-point $\mathbf{x}_i$ and its corresponding explanation $E(\mathbf{x}_i, f)$) and use case label ($f(\mathbf{x}_i)$). This input/output definition is used to generate data to train and evaluate the agent. In Section 5 we show that training an algorithmic agent using this framework allows us to corroborate findings about the usefulness of explanations as studied in Hase and Bansal [16].

1. A base dataset ($\mathcal{D} = \{(\mathbf{x}, y)\}$), on which the explanation method's utility will be evaluated.
2. The base model family, $\mathcal{F}$, which is the family of prediction models. The trained base model $f \in \mathcal{F}$, which is trained on $\mathcal{D}$, is used to generate explanations.
3. A function that defines a use case label, which is a ground truth label for the use case. This label corresponds to the correct answer for each question that a user must answer in the user study.

Now, we describe how to train and evaluate each agent model.[3] We introduce a three-step general framework: (1) data generation, (2) agent training, and (3) agent evaluation. Since the three steps of our framework closely parallel the steps in a standard ML workflow for model training and evaluation, canonical ML training techniques can also be used to train an algorithmic agent. The primary difference from the standard ML workflow is that in our framework, Step (1) is much more involved to ensure that the data generated as input to the algorithmic agent reflects the information content that would be presented to a human subject in a user study as shown in Figure 2.

**Step 1: Data Generation.** The goal of this step is to use the researcher's choice of information content and use-case-specific components to generate a dataset to train and evaluate the agent. The dataset consists of observation and use case label pairs, where the information content defines *what information* is included in each observation. We show a concrete example of data generation for the forward simulation use case in Figure 3. The specific construction of each observation can vary depending on several factors to accommodate a diverse range of use cases and user study designs:

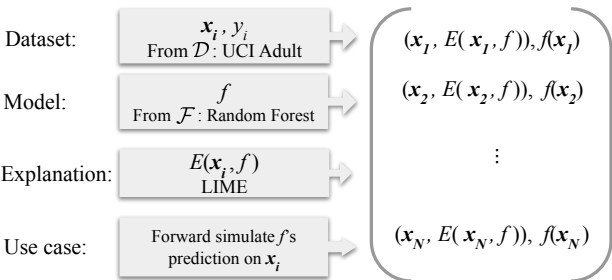

Figure 3: An example of the data generation process for the forward simulation use case from Hase and Bansal [16]. (Left) For this particular information content type, each observation contains the data-point $\mathbf{x}_i$ and the model explanation for that datapoint $E(f, \mathbf{x}_i)$. The three use-case-specific components are the base dataset $\mathcal{D}$, base model family $\mathcal{F}$, and function that defines a use case label. (Right) These components are used to generate a dataset of $N$ observations.

---

[3]Note the distinction between the *prediction model* being explained (which we call the "base model") and the separate *agent model*.

*Local vs. global.* The `SimEval` framework can be used to evaluate both local and global explanations: A local explanation method $E(f, \mathbf{x})$ takes as input prediction model $f$ and a single data-point $\mathbf{x} \in \mathcal{X}$. A global explanation method $E(f, \mathcal{D})$ takes as the input model $f$ and a dataset $\mathcal{D}$. The specific construction of each observation (i.e., whether the observation contains a single explanation vs. an aggregation of explanations) depends on whether the use case is a global problem (using explanations to answer a question about the entire model) or a local problem (using an explanation to answer a question about the model's behavior at a particular point). For global problems, the use case label is the *same* for a given model as opposed to for local problems, the use case label varies depending on the data-point of interest. A longer discussion can be found in the Appendix E.

*Multiple datasets $\mathcal{D}_i$ and models $f_i$.* Instead of building each observation using a single base dataset and base model, for some use cases we sample each observation $i$ from a *different* base dataset $\mathcal{D}_i$ (and use it to train a new base model $f_i$). We do so because (a) global problems require multiple models to have a diverse set of use case labels and (b) it may be undesirable for the agent to overfit to a single base dataset.

*Human Factors.* When selecting explanation methods for evaluation, it is important to consider factors that influence the way humans reason about explanations. We advocate for researchers to select explanations that also satisfy desirable proxy metrics such as faithfulness and stability. Additionally, it is important to select explanations that are suitable for human use. We discuss these considerations in the context of our experiments and user study in more detail in Appendix J.

**Step 2: Agent training.** After generating a dataset, the researcher trains an algorithmic agent (e.g., a machine learning model) to predict the use case label for a given observation. While this step can be viewed as equivalent to training an ML model (i.e., requiring standard hyperparameter search and tuning), this step is also analogous to "training" phases in human subject studies in which humans are provided with labeled examples to convey relevant knowledge for the use case (e.g., presented example explanations and expected conclusions). The training process allows the algorithmic agent to implicitly learn relevant domain knowledge about the use case from labeled examples, even when the researcher does not have any knowledge a priori of how to predict the use case labels.

In terms of model architecture selection, the researcher should select an agent model that is able to encode and learn meaningful representations of the observations that are input to the agent (which include the data-point and explanations). For example, the researcher needs to define the encoding for the choice of explanation. There are some standard ways to encode certain types of explanations (e.g., a feature attribution explanation for tabular data can be encoded as a vector of length $d$ if $\mathbf{x} \in \mathbb{R}^d$ and concatenated to the data-point to create an observation vector of length $2d$). However, in general, the researcher can leverage their domain knowledge in selecting the model: for example, a CNN architecture is appropriate for a base dataset containing images. We discuss in Appendix E how we encode the explanations evaluated in our experiments.

**Step 3: Agent evaluation.** After training the agent, the researcher evaluates the agent's ability to make predictions on unseen test examples. This phase is analogous to the evaluation portion of typical user studies, where humans are tested on new examples. The agent's accuracy on these examples is a measure of the predictiveness of the information content for the use case: a relative higher accuracy is evidence that a particular type of information content is more useful. We note that the agent's performance (i.e. test set accuracy) is *not* intended to be interpreted as the exact, anticipated human subject's performance when given the same set of information content as the algorithmic agent. We discuss several factors beyond the predictive information in the explanation that may affect human performance in Appendix J. As such, researchers should focus on *statistically significant differences* in agent performance when interpreting their SimEvals, as smaller differences between information content may not generalize to human performance.

## 4   Instantiating `SimEvals`

We instantiate our general framework to study the utility of popular post-hoc and by-design explanations (e.g., LIME, SHAP, Anchors, GAM) for three different use cases, spanning a diverse set of problems (forward simulation [16], counterfactual reasoning [25], and data bugs [20]). The first two use cases are local problems that focus on base model behavior at a specific point, while the third is a global problem that asks questions about the base model behavior more broadly. The human subject evaluations that studied these use cases primarily focused on studying *which explanation method* best helps users answer the use case questions.

Next, we focus on a key portion of the `SimEval` framework that must be carefully defined, discussing the data generation process in Step (1) for each use case. Further details on data generation algorithms and agent architectures used for each use case can be found in Appendices B and E.

**Forward simulation** measures whether a user correctly simulates (i.e., predicts) the ML model's prediction on a given data-point [35, 8, 16]. Our particular set-up is motivated by Hase and Bansal [16], where the authors test whether providing users with explanations helps users better predict a base model's behavior when compared to a control setting in which users are not given explanations. Here, each observation given as input to the `SimEval` agent consists of a single data-point and explanation $(\mathbf{x}_i, E(\mathbf{x}_i, f))$, where $\mathbf{x}_i$ is sampled from the base dataset $\mathcal{D}$. Following the same procedure as Hase and Bansal, we split $\mathcal{D}$ into train, test, and validation sets. The agent is trained on a dataset of observations from the training set and is then evaluated on observations from the unseen validation set. The use case label $u_i = f(\mathbf{x}_i)$ is the model $f$'s output.

**Counterfactual reasoning** is a concept from philosophy used to describe "what if" conditionals that are contrary to fact [43]. Counterfactual reasoning can enable individuals to reason about how to achieve a desired outcome from a prediction model. We study a specific instantiation of counterfactual reasoning from Kumar et al. [25]. Our use case examines whether a model's output $f(\mathbf{x}_i)$ increases for some input $\mathbf{x}_i$ in a counterfactual setting where a given feature of $\mathbf{x}_i$ is increased. Like forward simulation, each observation used to train and evaluate the agent consists of a single data-point but differs in that each observation also includes the base model prediction on that data-point: $(\mathbf{x}_i, f(\mathbf{x}_i), E(\mathbf{x}_i, f))$. For a given input $\mathbf{x}_i$, the use case label $u_i$ answers the question: "Does $f(\mathbf{x}_i)$ increase if a particular feature of $\mathbf{x}_i$ is increased?". We use the same agent model architecture as in the forward simulation use case.

One important difference for counterfactual reasoning is that each base dataset $\mathcal{D}_i$ and its corresponding base model $f_i$ can differ across observations. We construct the agent's observations in this way to prevent the agent from simply memorizing where an individual base model is increasing versus decreasing with respect to a particular feature, instead encouraging the agent to learn a heuristic for this use case that generalizes across different base models (further discussion provided in Appendix D).

**Data bugs** is a use case where the user is asked to predict whether the base model was trained on a dataset containing a systematic error. For this use case, the researcher additionally needs to provide a "bug definition" $B$, which corrupts the given base dataset. We consider bug definitions studied in Kaur et al. [20] and summarize their motivation for identifying each of these bugs: The *missing value* bug occurs when missing values for a feature are imputed using the mean feature value. This can bias the prediction model towards the mean value. The *redundant feature* bug occurs when the dataset contains features that are redundant, or contain the same predictive information. This can distribute the true importance of the measure between the redundant features, failing to convey the measure's actual importance. We note that while explanations may not be the only approach to identify these data bugs, the authors demonstrate that model explanations *can* help users with this use case.

A distinguishing feature of the data bugs use case is that the study participants in Kaur et al. [20] were given a *set* of data-points and explanations per observation because it might be easier to detect a model bug if given more than one data-point (an illustration provided in the Appendix F.3). As such, each observation for the agent is also a set of data-point, model prediction, and explanation tuples: $\{(\mathbf{x}_i^{(j)}, f(\mathbf{x}_i^{(j)}), E(\mathbf{x}_i^{(j)}, f_i))\}_{j=1}^S$, where $\mathbf{x}_i^{(j)}$ denotes the $j$th element in the set which is part of the $i$th observation. In our experiments, we vary the proportion of datasets $\mathcal{D}_i$ that have bug $B$. For the datasets that have the bug, we set $u_i = 1$ and $u_i = 0$ otherwise.

For data bugs, each base dataset $\mathcal{D}_i$ and its corresponding base model $f_i$ differ across observations because data bugs is a *global problem*, i.e. the use case label $u_i$ is a property of base model $f_i$. Specifically, we derive many $\mathcal{D}_i$ by subsampling one dataset $\mathcal{D}$. This allows us to create a variety of training (and evaluation) examples for the agent with varying use case labels, encouraging the agent to learn a heuristic that generalizes better across different sets of points.

## 5   Comparing Simulation and Humans

To demonstrate that `SimEvals` are able to identify promising explanations, we run `SimEvals` on the three use cases discussed in Section 4 and compare the algorithmic agent results to user study results, which come from either prior user studies or our own Mechanical Turk studies. Specifically,

we validate that the algorithmic `SimEval` agent performs relatively well when given an explanation that a human can perform well with, and performs relatively worse when given an explanation that is not useful to a human. We include training details for our `SimEvals` in Appendix F, extended results (with error bars) and comparisons with human results in Appendix G.

## 5.1 User study details

In our experiments, we compare the algorithmic `SimEval` agent's performance to the average human subject performance at predicting the use case label in a user study. For the **Forward Simulation** use case, we compare the algorithmic agent's performance to human performance as reported in a previously conducted independent user study by Hase and Bansal [16]. We conduct new user studies to measure human performance on two of the use cases. For **Counterfactual Reasoning**, there is no existing user study from the motivating work from Kumar et al. [25], necessitating that we run our own. For **Data Bugs**, we conducted our own study that, unlike the original one by Kaur et al [20], individually measures the usefulness of candidate explanations. We expand on the differences between the two studies in Section 5.4.

Our user studies were conducted on Amazon Mechanical Turk, where we presented Turkers with the observations sampled from the dataset that `SimEvals` were trained and evaluated on. This study was approved by the IRB (STUDY2021_00000286) for exempt Category 3 research. We choose a between-subjects study design, where each participant is randomly assigned to one explanation settings from the `SimEvals`. In non-baseline settings, participants are also provided with explanations for each $\mathbf{x}_i$, which we described to the Turkers as importance scores (Figure 7). The study is split into two parts. First, each participant completes a "Train" survey (analogous to agent training) where they are asked to predict the use case label for 15 observations and provided feedback on whether their responses were correct after every 5 observations. Then participants complete a "Test" survey (analogous to agent evaluation) where they predict the use case label for another 15 observations without feedback. Each question corresponds directly to 1 randomly sampled observation used to train or test the algorithmic agent. For each use case, we recruit 80 total participants: 20 participants for each explanation setting. Other details, including the study interface and instructions for each use case, participant recruitment process, details of initial pilot studies are provided in Appendix H.

## 5.2 Forward Simulation

Following Hase and Bansal [16], we compare the prediction accuracy of our agent in the 'No explanation' baseline setting (where the agent/human receives only the data-point without explanation) with its prediction accuracy when given both the data-point and its corresponding explanation. The authors evaluate 5 explanations in their user study: 2 are well-known explanations (LIME [37], Anchors [38]) and the rest are bespoke. In this work, we do not evaluate the authors' bespoke methods since none outperformed the standard methods as described in Appendix C.

Table 1: (Left) Test accuracy of the agent model (averaged across 3 random seeds) when varying the number of training observations the agent receives to perform forward simulation, recreating trends in average human subject test accuracy (Right) from Hase and Bansal [16].

| | Agent Test Accuracy (Varying Train Set Size) | | | | Human Test Accuracy from [16] |
|---|---|---|---|---|---|
| Explanation | 16 | 32 | 100 | 1000 | |
| LIME | 94.2% | 99.8% | 100% | 100% | 81.9% ± 8.8% |
| Anchors | 89.2% | 93.5% | 94.7% | 93.7% | 75.8% ± 8.5% |
| No explanation | 82.3% | 83.7% | 85.7% | 88.7% | 70.7% ± 6.9% |

*Findings:* Hase and Bansal [16] found that participants who were given the LIME explanation for forward simulation achieved an accuracy increase that is more than twice that achieved by other methods. As shown in Table 1, we find that `SimEvals` corroborates this result from the user study even when varying the agent's train set size. In Appendix G.1, we provide additional `SimEvals` which hypothesize that SHAP and GAM are also promising choices for this use case.

## 5.3 Counterfactual reasoning

The counterfactual reasoning use case is introduced by Kumar et al. [25], where the authors discuss why SHAP feature importance scores [29] do not explicitly attempt to provide guidance for the use case. Specifically, a positive SHAP feature importance score does not necessarily imply that increasing that feature will increase the model's prediction. We additionally hypothesize that an approximation-based explanation like LIME might be more useful for this use case than SHAP. We conduct a user study for this use case since Kumar et al. did not conduct a user study.

Table 2: Average agent test set accuracy (Left) aligns with average Turker test survey accuracy (Right) for the counterfactual reasoning use case. (Left) Agent test set accuracies as we vary the number of training observations the agent receives to perform counterfactual reasoning. (Right) The 95% confidence interval of average user accuracy on 15 test observations where for each explanation setting we recruited $N = 20$ Turkers.

| | Agent Test Accuracy (Varying Train Set Size) | | | | Human Test Accuracy |
|---|---|---|---|---|---|
| Explanation | 4 | 16 | 100 | 1000 | |
| LIME | 92.9% | 94.5% | 99.2% | 99.7% | 69.4% ± 12.4% |
| SHAP | 56.3% | 52.5% | 57.3% | 64.9% | 41.4% ± 6.14% |
| GAM | 56.0% | 53.5% | 57.3% | 63.5% | 45.7% ± 7.19% |
| No explanation | 52.1% | 55.6% | 57.9% | 60.3% | 48.6% ± 5.61% |

*Findings:* As shown in Table 2 (Left), we find that the simulated agent had significantly higher accuracy when given LIME explanations. Additionally, we find that this trend is consistent across all training set sizes: the agent achieves a 37% increase in validation accuracy when given LIME instead of any other explanation after observing only 4 train observations. These results suggest that, of the set of explanations considered, LIME is the most promising explanation for the counterfactual reasoning use case. In Appendix G.5 we demonstrate that LIME outperforms all other explanation methods consistently across different types of model architectures.

As shown in Table 2 (Right), the gap in agent performance between LIME and the other explanations is reflected in the human test accuracy in our Turk study. In particular, a number of Turkers are able to use the predictiveness of LIME to complete the use case with high accuracy; whereas, in general, Turkers using SHAP, GAM, and No Explanation perform no better than random guessing. We ran an ANOVA test and found a statistically significant difference between the Turkers' accuracy using LIME vs. all other explanation settings and no statistically significant difference between the other settings (Appendix G.2). Note that there is also significant overlap in the error bars of agent performance on SHAP, GAM, No Explanation (Table 6). We observe greater variance in participant accuracy between participants for LIME compared to the other explanations. We believe that this is because many, but not all, Turkers were able to learn how to use the LIME explanation successfully.

## 5.4 Data bugs

Kaur et al. [20] study two explanations (SHAP [29] and GAMs [17]) on two types of data bugs (missing values and redundant features). In their study, users were prompted to suggest bugs that they believed may be present via semi-structured explorations using these explanation tools. They found that 4 out of 11 users mentioned a missing values bug and 3 users mentioned redundant features. Their results suggest that both SHAP and GAM may be useful for finding these types of data bugs, though they do not individually test each explanation's usefulness and they also did not consider providing users with baseline information content (without model explanations) to compare with settings where users were presented with explanations. Thus, for this use case, we conducted our own Turk user study on the missing values bug, evaluating both the explanations considered in the original study and including additional explanation/baseline comparisons. We present `SimEval` results for the missing values bug below (redundant features can be found in Appendix G.4).

**Missing Values:** Kaur et al. [20] implement the missing values bug by replacing the 'Age' value with the mean value of 38 for 10% of adults with $> 50k$ income. We detail how we recreate this bug in Appendix F.3. In addition to comparing various explanation methods, we also use `SimEvals` to evaluate the effect of varying the number of data-points per train set observation (e.g. the size $S$

of each set). We note is another type of information content that the researcher must select when running a user study.

Table 3: Average agent test set accuracy (Left) aligns with average Turker test survey accuracy (Right) for the data bugs (missing values) use case. (Left) Agent test set accuracies as we vary the number of training observations the agent receives to perform counterfactual reasoning. (Right) The 95% confidence interval of average user accuracy on 15 test observations where for each explanation setting we recruited $N = 20$ Turkers. We denote explanation settings that were considered in the original user study with $\star$.

| | Agent Test Accuracy | | | | Human Test |
| | (Number of Data-points Per Observation) | | | | Accuracy |
| Explanation | 1 | 10 | 100 | 1000 | |
|---|---|---|---|---|---|
| SHAP $\star$ | 63.2% | 84% | 99.8% | 100% | 67.4% $\pm$ 11.5% |
| GAM $\star$ | 64.8% | 87.7% | 100% | 100% | 64.4% $\pm$ 7.2% |
| LIME | 55.2% | 56.9% | 64.5% | 75.9% | 48.0% $\pm$ 5.4% |
| Model Prediction | 57.5% | 57.4% | 58.2% | 67.3% | 40.7% $\pm$ 5.2% |

*Findings:* As shown in Table 3 (Left), we verify using `SimEvals` that SHAP and GAM explanations contain predictive information to identify the missing values bug. We also find that the accuracy of the algorithmic agent drops when given too few data-points (i.e., for observation sets of size $\leq 10$). This means that a small number of data-points do not contain sufficient signal to detect the bug. Since the researcher would need to present a large number of data-points and explanations to a human user in a human-interpretable manner, a user interface is needed where the explanations are presented in an aggregated fashion, which is indeed the approach taken by both Kaur et al. [20] and our study. In Appendix G, we observe similar trends in the relative performance of explanation methods even when varying the bug strength and the parameterization of the agent model.

In our Turk study, we utilized the findings from our `SimEvals` experiments and presented the explanations for $S = 1000$ data-points per observation in an aggregated fashion (via scatter plots as shown in Figure 11). As shown in Table 3 (Right), participants assigned to SHAP or GAM consistently outperformed the participants assigned to LIME or the Model Prediction baseline, even though our participant population for the user study were dissimilar from the participant population (data scientists) of the original user study. We conducted pairwise statistical analyses and found statistical significance between SHAP and LIME/Model Prediction as well as GAM and LIME/Model Prediction and no statistical significance between other pairs of experimental conditions.

## 6 Discussion

Our experiments and analyses in Section 5 demonstrate that `SimEvals` are able to provide an informative measure of the predictive information contained in a design set-up, and as such can identify promising explanations from a set of candidate explanations. However, we emphasize that because `SimEvals` only measure predictive information, the algorithmic agent's accuracy *cannot* be directly interpreted as the explanation's utility to a human, as evidenced by the differences between raw agent and human test accuracies in Tables 1, 2, and 3. Furthermore, we note that small differences between `SimEval` agent's relative performance on explanations (e.g., in Table 2) may not generalize to humans. We discuss potential human factors that may contribute to these differences between `SimEval` and human performance in Appendix J. These findings suggest an important direction for future research to better understand these differences between agent and human, i.e., how human factors beyond predictive information affect humans' ability to reason about model explanations.

## 7 Conclusion

We proposed a use-case-grounded algorithmic evaluation called `SimEvals` to efficiently screen choices of information content and identify good candidates for a user study. Each `SimEval` involves training an agent to predict the use case label given the information content that would be presented to a user. We instantiated `SimEvals` on three use cases motivated by prior work, and demonstrated that the agent's test set accuracy can be used as a measure of the predictiveness of information content (more specifically, of an explanation method) for each use case. We found that humans perform significantly better on explanations that `SimEvals` selects as promising compared to other explanations. While our experiments focus primarily on the design choice of selecting explanation

methods, we note that researchers can also use `SimEvals` to screen the predictiveness of other types of information content, such as hyperparameter selection for explanation methods or the baseline information given with each observation. We hope that this work can be incorporated into future explanation evaluation workflows to enhance the efficiency and effectiveness of user study design.

# 8    Acknowledgements

We are grateful for helpful feedback from Chirag Agarwal, Zana Bucinca, Alex Cabrera, Lucio Dery, Elena Glassman, Joon Sik Kim, Satyapriya Krishna, Isaac Lage, Martin Pawelcyzk, Danish Pruthi, Rattana Pukdee, Isha Puri, Junhong Shen, Manuela Veloso.

This work was supported in part by the National Science Foundation grants IIS1705121, IIS1838017, IIS2046613, IIS2112471, an Amazon Web Services Award, a Facebook Faculty Research Award, funding from Booz Allen Hamilton Inc., and a Block Center Grant. Any opinions, findings and conclusions or recommendations expressed in this material are those of the author(s) and do not necessarily reflect the views of any of these funding agencies.

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
