# A   Examples of algorithmic agents in prior work

We can unify existing ad-hoc examples that use algorithmic agents to perform use case grounded evaluations under our framework. We group existing algorithmic evaluations into two categories: Heuristic-Based and Learning-Based.

## A.1   Heuristic-Based Evaluations.

These algorithmic agents were not trained, and instead were defined using hand-designed heuristics. In contrast, our agent is trained on observations and use case labels.

**Expo** [33]: The use case studied is whether an explanation can be used to determine how to modify features of an instance to achieve a target prediction. There is no agent training process: the agent uses a randomized greedy heuristic (which involves picking the largest feature). The agent is evaluated by the number of modifications that it makes to reach the target prediction.

**Influence functions** [23] **/ RPS** [45]: The use case studied is whether explanations can be used to identify mislabeled examples in a dataset. The agent is also heuristic-based, flagging points with the largest influences to be inspected for potentially being mislabeled. The agent is evaluated by the number of correct points it selects for inspection.

**Anchors** [38]: The use case studied is whether explanations can help a user make accurate predictions on unseen instances. The authors specify a heuristic that they use to compare Anchors with LIME. The heuristic for Anchors is simply to check whether the conditions are satisfied. The heuristic for LIME is to use the linear approximation directly (while additionally checking whether the point of prediction is near the approximation point). The agent is evaluated using its precision when used to predict labels on unseen instances.

## A.2   Learning-Based Evaluations.

**Student-teacher Models** [36]: The use case studied is whether explanations can help a student model learn to simulate a teacher model for a sentiment analysis task and question answering task. The authors quantify the benefit of an explanation as the improvement in simulation accuracy when the agent has access to explanations during the training process versus when the agent does not. One important distinction from our approach is that it is noted in [36] that their algorithmic evaluation does *not* attempt to replicate the evaluation protocols used in a human subject study. This point is affirmed by the significant difference between the results of the authors' human subject study vs. their algorithmic evaluation. In contrast, our approach is designed with a human subject study in mind: we explicitly do attempt to construct the agent's observations in a way that reflects the information that would be presented to a human subject.

Other work [13, 42] similar to [36] that evaluates explanations without a human in the loop typically examines how explanations can improve the model training process. In contrast, our work focuses on use cases where explanations are intended to be shown to and used by humans.

## A.3   Distinction from prior work

In Table A.3, we summarize prior work along two main axes: their algorithmic evaluation of explanations (if they conducted them) and their human evaluation of explanations (if they conducted them). On the algorithmic evaluation front, no prior work allows their agent to learn how to use explanations (i.e., a researcher does not need to specify a heuristic for how the explanation will be used) *and* provides a general framework that encompasses many use cases. Furthermore, no prior work extensively verifies their evaluation framework with human evaluation to show the comparability between algorithmic and human performance, whereas our work evaluates as many explanations as many user studies.

# B   Use Case Algorithms

We provide more detailed algorithms to describe how the dataset is generated for the forward simulation and counterfactual reasoning use cases.

| | Algorithmic Evaluation | | Human Evaluation | | |
|---|---|---|---|---|---|
| | Learning-based? (i.e.,no heuristic required). | Does framework generalize? | Conducts a user study? | Agent matches human? | Num. explanations (baselines) evaluated |
| Expo | No | No | Yes | No | 1 (1) |
| IF/RPS | No | No | No | – | – |
| S-T Models | Yes | No | Yes | No | 2 (1) |
| Anchors | No | No | Yes | Yes | 2 (0) |
| User studies | – | – | Yes | – | $\sim$2-4 (1-2) |
| **Ours (SimEvals)** | Yes | Yes | Yes | Yes | 4 (2) |

## B.1 Forward simulation

- Given base dataset $\mathcal{D}$, split into train/test/validation
- Train model $f$ on the train set
- Let $N_T$ be size of desired training dataset and $N_V$ be size of the desired validation dataset
- Sample $N_T$ points from the train set and for each point $\mathbf{x}_i \in \mathcal{D}_{train}$ generate an explanation of $f$ at that point, add the tuple $(\mathbf{x}_i, E(\mathbf{x}_i, f))$ to the training dataset.
- Repeat process for $N_V$ points but from the validation set

Note the baseline condition would be generated in the same way, but excluding the explanation.

## B.2 Counterfactual Reasoning

- Given base model family $\mathcal{F}$, desired dataset size $N$
- For $i = 1, ..., N$:
    - Sample saddle-point function $g_i$ as detailed in Appendix D.2
    - Train $f_i$ on $g_i$ as detailed in Appendix D.2
    - Sample new point $\mathbf{x}_i$ from $g_i$
    - Get prediction $f_i(\mathbf{x}_i)$
    - Generate $E(\mathbf{x}_i, f_i)$ of $f_i$ at $\mathbf{x}_i$
- Set the $i$th observation to be: $(\mathbf{x}_i, f_i(\mathbf{x}_i), E(\mathbf{x}_i, f_i))$

## B.3 Data Bugs

---

**Algorithm 1** generates a dataset of $N$ observations for the data bug use case.

---

1: Given base dataset $\mathcal{D}$, base model family $\mathcal{F}$ and bug specification $B$
2: Let $N$ be the size of desired dataset,
    $S$ be the observation set size
3: **for** $i$ in 1...$N$ **do**
4:    Subsample $\mathcal{D}_i$ from $\mathcal{D}$
5:    With probability 0.5, apply $B$ to $\mathcal{D}_i$
6:    Train $f_i$ on $\mathcal{D}_i$
7:    **for** $j$ in 1...$S$ **do**
8:        Sample $\mathbf{x}_i^{(j)}$ from $\mathcal{D}_i$
9:        Get prediction $f_i(\mathbf{x}_i^{(j)})$
10:       Generate $E(\mathbf{x}_i^{(j)}, f_i)$ of $f_i$ at $\mathbf{x}_i^{(j)}$
11:    **end for**
12:    Set the $i$th observation to be:
        $\{(\mathbf{x}_i^{(j)}, f_i(\mathbf{x}_i^{(j)}), E(\mathbf{x}_i^{(j)}, f_i))\}_{j=1}^{S}$
13: **end for**

---

## C   Summary of prior user studies

We discuss the findings from the prior user studies that we recreate results from.

**Forward Simulation** from [16]. The authors evaluate 5 explanations in this paper: 2 (LIME, Anchors) are well-known, open-source explanations and 3 (Prototype, DB, and Composite) are bespoke explanations. In the tabular setting, the authors find that LIME outperforms all other explanations (change of 11.25% in user accuracy compared to Anchors 5.01%, Prototype 1.68%, 5.27%, 0.33%). The base accuracy before explanations was 70.74%. Since none of the bespoke explanations outperformed the open-source explanations, we chose not to re-implement them in our study.

**Data Bugs** from [20]. The authors evaluate 2 explanations: SHAP and GAM. The set-up of this study did not explicitly ask the users to answer whether the bug was present, but rather asked the user to identify the bugs themselves (i.e., if a user could identify the existence of the bug, then the explanation was useful). The authors find that 4 out of 11 data scientists were able to identify the missing values bug and 3 out of 11 the redundant features bug. While the authors do not distinguish when SHAP is helpful versus GAM. From Figure 1 of their paper, SHAP and GAM seem to have similar visualizations and thus might equally help a user. We find similar results with our simulated agent.

## D   Counterfactual Reasoning Use Case Details

### D.1   Distinction from Counterfactual Explanations

There are significant differences between our use case and the desiderata for which counterfactual explanation methods for ML models are developed.

**Why existing counterfactual explanation methods do not apply:** A growing body of IML research has proposed new methods to provide counterfactual explanations for a black-box model [41, 31, 19]. These methods provide explanations that fulfill the desiderata introduced in [43], defined formally as:

**Definition D.1.** Given input $\mathbf{x}$, model $f$, and desired outcome set $\mathcal{Y}$, counterfactual explanation $\mathbf{x}'$ is the *closest possible point* to $\mathbf{x}$ such that the model's prediction on the counterfactual point is in the desired outcome set: $f(\mathbf{x}') \in \mathcal{Y}$.

The real-world utility of counterfactual explanations relies on a large number of assumptions [6]. Many variants of the counterfactual problem have been proposed to account for a wide range of real world objectives and constraints, such as the feasibility or difficulty of the proposed actions (recourse) [41]. Despite these varying goals, however, to our knowledge prior work on counterfactual explanations shares the general problem set-up in Definition D.1.

While this problem set-up in Definition D.1 is *related* to our use case, it is significantly *different* for a number of reasons. Primarily, while the counterfactual explanation $\mathbf{x}'$ provided by these methods is encouraged to be "close" to the original input $\mathbf{x}$, the counterfactual $\mathbf{x}'$ can increase or decrease any subset of input features within the problem's constraints. Thus, there may be a wide range of possible counterfactuals $\mathbf{x}'$ of varying "closeness" to $\mathbf{x}$ that result in the desired outcome $f(\mathbf{x}')$. In contrast, the use case asks a more specific question of what happens when a specific *feature $i$* is *increased* and all other features are held constant.

In a scenario where it may be possible to achieve an increased model prediction $f(\mathbf{x}')$ by changing other subsets of features excluding feature $i$, then the provided counterfactual $\mathbf{x}'$ will give no information relevant to the use case. Similarly, it may be possible that $f(\mathbf{x}')$ may decrease when some feature $j \neq i$ is changed *and* feature $i$ is increased, suggesting that $f$ decreases when feature $i$ is increased; but if all other features $j \neq i$ are held *constant*, then $f$ *increases* when feature $i$ is increased. In these cases, the optimal counterfactual explanation $\mathbf{x}'$ may give no information about how $f(\mathbf{x})$ varies with feature $i$.

Due to this misalignment between the goals of existing counterfactual explanation methods and our particular use case, we exclude these counterfactual explanation methods from our evaluation. We instead focus our evaluation on common feature attribution methods.

## D.2 Synthetic Data

Inspired by Kumar et al. [25], we construct a 2D toy base dataset comprising of points sampled from a saddle-point function. We construct a low-dimensional dataset to reduce the complexity of the task given to MTurkers, as prior work [35] found that Turkers achieved higher accuracy when presented with data that had fewer features.

Each data-point $(\mathbf{x}_i : [x_{i,1}, x_{i,2}], y_i)$ used to train predictive model $f_i$ is sampled from its *own* saddle-point function $g_i$ with parameters chosen stochastically (detailed further in Section D.2.1 below). Because each observation is sampled from a different function, each observation also has its own predictive model $f_i$. Importantly, we chose to sample points from a saddle-point function (with no noise) because saddle-point functions are either *strictly concave* or *strictly convex* with respect to each feature $x_{i,1}, x_{i,2}$. Thus, if the predictive model $f_i$ correctly learns the saddle-point function $g_i$, then "ground truth" use case labels $u_i$ of whether or not the predictive model's output increases with feature $x_{i,2}$ are available for all points.

**Why each point from a separate function?:** We chose to sample every observation given to the agent from its own separate saddle-point function to increase the difficulty of the agent prediction task. Consider for comparison an alternative naive dataset generation procedure where all observations $\{(\mathbf{x}_i, f(\mathbf{x}_i)), u_i\}_{i=1}^N$ are sampled from the *same* saddle-point function $g$ (and consequently share the same predictive model $f$). Given enough observations $N$, because $f$ is smooth and either strictly concave or convex, an agent can memorize for which points $(\mathbf{x}_i, f(\mathbf{x}_i))$ the function $f$ is increasing vs. decreasing with respect to feature $x_{i,2}$. Thus, the agent can effectively memorize the use case labels without needing to use explanations. This is undesirable as our intention in training the agent is to evaluate candidate explanation methods.

Therefore instead of using the same predictive model for all observations, we trained different predictive models on observations sampled from different saddle-point functions to (a) prevent the agent from learning a heuristic specific to any singular function and (b) to encourage the agent to learn a heuristic that will generalize well on new observations from different functions.

Below, we detail how data-points and use case labels are defined for each saddle-point function (Section D.2.1) and how the agent's train and validation sets are constructed (Section D.2.2).

### D.2.1 Saddle-Point Functions

For each observation $i$, we construct saddle-point function $g_i$ as follows:

First, critical points $x_1^*, x_2^*$ are chosen uniformly at random in range $[10, 15]$. Second, an indicator variable $Z$ is sampled from a Bernoulli distribution with parameter $p = 0.5$.

We define saddle-point function $g_i$ as:

$$g_i(x_1, x_2) = (x_1 - x_1^*)^2 - (x_2 - x_2^*)^2 \tag{1}$$

To sample data-point pairs $(\mathbf{x}_i : [x_{i,1}, x_{i,2}], y_i)$ and their corresponding ground truth use case labels $u_i$ from $g_i$:

- $x_{i,1}$ and $x_{i,2}$ are sampled uniformly in the ranges $[x_1^* \pm 5]$ and $[x_2^* \pm 5]$ respectively.

- Generate outcomes $y_i$ as:

$$y_i = \begin{cases} g_i(x_{i,1}, x_{i,2}) + 30, & \text{if } Z = 1 \\ -g_i(x_{i,1}, x_{i,2}) + 30, & \text{if } Z = 0 \end{cases} \tag{2}$$

- Generate use case labels $u_i$ using indicator variables $\mathbb{1}(\cdot)$ as:

$$u_i = \begin{cases} \mathbb{1}(x_{i,2} \leq x_2^*), & \text{if } Z = 1 \\ \mathbb{1}(x_{i,2} > x_2^*), & \text{if } Z = 0 \end{cases} \tag{3}$$

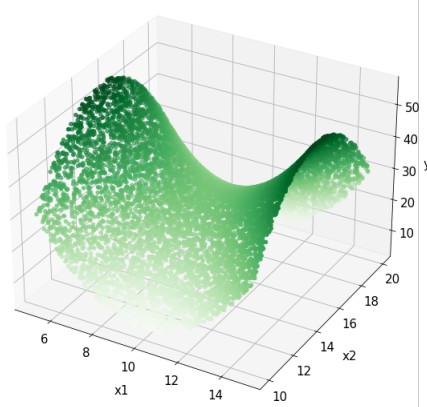

Figure 4: Scatterplot of $10,000$ points sampled from the saddle-plot function with $x_1^* = 10$, $x_2^* = 15$, and $Z = 1$. This function is *strictly concave* with respect to feature $x_2$.

Figure 4 shows the 2D toy dataset generated using critical points $x_1^* = 10$, $x_2^* = 15$ and $Z = 1$. Notice that because $y = g_i(x_1, x_2) + 30$ is strictly concave with respect to feature $x_2$, the function's output *increases* as $x_2$ increases for all points where $x_2 \le x_2^*$. At the critical point of $x_2^*$, the use case labels $u$ switch from 1 to 0: for points where $x_2 > x_2^*$, then the function's output *decreases* as $x_2$ increases.

In contrast, when $Z = 0$, then data-points $y = -g_i(x_1, x_2) + 30$ are generated from a function that is the negation of Figure 4 that is strictly convex with respect to feature $x_2$. Thus the use case labels are the negation of the use case labels for $Z = 1$: they are now 1 when $x_2 > x_2^*$.

### D.2.2 Running the Simulated Evaluation

Below we detail how predictive models $f_i$ and the agent's train and validation sets are constructed.

**Training predictive models $f_i$.** For each observation $i$, we generate $N = 5000$ data-points in range $[x_1^* \pm 5, x_2^* \pm 5]$ following Section D.2.1 and train the Light-GBM Regressor $f_i$. We validate by visual inspection that the learned models $f_i$ are "smooth", i.e. that the use case labels $u_i$ derived using the definition in Section D.2.1 are correct by visualizing the model's output $f_i(\mathbf{x}_i)$ over the domain $\mathbf{x}_i \in [x_1^* \pm 5, x_2^* \pm 5]$.

**Sampling observations $(\mathbf{x}_i, f_i(\mathbf{x}_i), u_i)$ to give to the agent.** After a predictive model $f_i$ has been trained, we sample a single point $\mathbf{x}_i$ from the fixed range $[11.25, 13.75]$ to construct observation $i$ for the agent. While this range is a *subset* of the range in which the critical points $x_1^*, x_2^*$ and the training data points were sampled, we intentionally sample the input points $\mathbf{x}_i$ in a more narrow range to increase the difficulty of the agent prediction task. Specifically, for the chosen range, the probability that the randomly selected critical point $x_2^*$ lies outside of this range is $50\%$, making it difficult for the agent to easily infer the use case label from the model prediction $f_i(\mathbf{x}_i)$ and covariates $\mathbf{x}_i$ alone.

Repeating this process several times, we construct a dataset of $10,000$ different observations (e.g. we train $10,000$ different models $f_i$ corresponding to $10,000$ different saddle-point functions $g_i$). For the 4 different explanation settings, we use each observation's respective trained predictive model $f_i$ to generate explanations. To train the agent, we split these $10,000$ observations into a train, test, and validation set.

### D.3 Limitations & Future Work

We note that there are a few limitations to our use case definition and evaluation. First, our use case definition does not account for the feasibility or cost of increasing the specific feature under study, which may be of practical importance in a real-world application. Second, we note that it may be difficult or subjective to derive ground-truth use case labels of whether or not a predictive model's outcome will increase locally if a feature is increased for a non-convex prediction model $f$. The naive approach of taking the gradient of $f$ at point $\mathbf{x}$ may not be representative of the behavior of function $f$ in a region around $\mathbf{x}$ [40]. In practice, we recommend that ground-truth use case labels

are defined empirically by averaging the model's predictions in a local region around input $\mathbf{x}$ using expert knowledge to define a meaningful and representative local region. Third, we note that to our knowledge, there is no explanation method that is explicitly developed to address this use case, which is an open direction for future work.

# E   Explanation Encoding

## E.1   Global vs. Local

The number of data-points included in each observation can vary depending on the *explanation type* (global vs. local explanations) and *problem type* (global vs. local problems). Table 4 summarizes how we recommend constructing each observation for each combination of problem and explanation type. We elaborate on several of our choices below.

Table 4: Describes how to construct each observation $i$ using global and local explanations for both global and local problems. Notation: $S$ is the size of the observation set; $\mathcal{D}$ denotes the base dataset; $f$ denotes the base model; $\mathbf{x}_i$ denotes a point sampled from the base dataset. Note that each observation $i$ for global problems by definition must use a *different* dataset $\mathcal{D}_i$ and model $f_i$; vs. local explanations may use the same model $f$ and dataset $\mathcal{D}$ (elaborated on below).

|  | Global Problem | Local Problem |
|---|---|---|
| Global exp | $E(f_i, \mathcal{D}_i)$ | $E(f, \mathcal{D})$ |
| Local exp | $\{E(\mathbf{x}_i^{(j)}, f_i)\}_{j=1}^S$ for $\mathbf{x}_i^{(j)}$ in $\mathcal{D}_i$ | $E(\mathbf{x}_i, f)$ |

- **Global Problem + Global Explanation:**   As discussed in Section 3, global problems require that each observation uses a different base predictive model $f_i$ because the use case $u_i$ labels are a property of the model (and thus would be the same for all observations if they all shared the same model). Therefore each observation $i$ contains the global explanation for that model $f_i$.

- **Global Problem + Local Explanation:**   Global problems require that each observation uses a different predictive model, but local explanations only explain the model's prediction on a single data-point. A single data-point may not contain enough information to infer a global property of predictive model $f_i$. To address this problem, the agent can take *multiple* (e.g. a set of $S$) data-points and explanations per observation.

- **Local Problem + Global Explanation:**   A local problem defines a different use-case label for each individual data-point $\mathbf{x}_i$. However, each base model $f$ and base dataset $\mathcal{D}$ only has 1 global explanation $E(f, \mathcal{D})$. As such, if the *same* model $f$ and dataset $\mathcal{D}$ are used for all observations $i$ (as in the Forward Simulation use case introduced in the main text, then the global explanation $E(f, \mathcal{D})$ will be constant across all observations $i$. We note that a researcher can choose to vary the predictive model $f_i$ and dataset $\mathcal{D}_i$ across observations $i$ (as in the Counterfactual Reasoning use case introduced in the main text; in this case the global explanation $E(f_i, \mathcal{D}_i)$ will vary across observations.

- **Local Problem + Local Explanation:**   Like the previous example, the predictive base model can vary or be held constant across different observations $i$. Each observation contains the base model's explanation on sampled point $\mathbf{x}_i$.

## E.2   Encoding explanations in our experiments

The explanations we consider in our experiment are LIME, SHAP, Anchors, and GAM:

- LIME and SHAP are both local explanations that return an importance score for each feature. We encode each explanation as a vector of length $d$ if $\mathbf{x} \in \mathbb{R}^d$ and concatenate the explanation to the data-point to create an observation vector of length $2d$. Note that in Forward Simulation, to recreate the original setting, we extend the vector of length $d$ with additional information from the LIME package as discussed in Appendix G.1.

- To get the GAM explanation, we used method `explain_local` out-of-the-box from the InterpretML package [32].

- To get the Anchors explanation, we call the Anchors implementation [38] which returns a set of anchors for a given data-point. For categorical features, the anchor is one of the values that the feature can take on. For continuous features, the anchor can take the form of a lower and upper bound. We include both the upper and lower bound for each continuous feature in the observation. For a data-point $\mathbf{x}$ with $d_{cat}$ categorical and $d_{cont}$ continuous features, the final observation has dimension at most $2d_{cat} + 3d_{cont}$.

### E.3 Encoding other kinds of explanations

There are other types of explanations that we do not consider in this work, but could also be studied using our proposed framework. We discuss some potential ways to encode these other types of explanations:

- Saliency maps: These types of explanations provide a "heat map" over an image input, where larger values signify a larger contribution of that pixel to the model's prediction. Suppose the dimensions of the input image are $d \times d \times c$ and the dimensions of the saliency map are $d \times d \times k$, one could define the observation as a $d \times d \times c + k$ input and instantiate the agent as a convolutional network. In Appendix I, we use this set up to replicate results for an image user study. Note that these types of explanations will most likely require a neural network-based agent architecture.

- Counterfactual explanations: These types of explanations provide the closest possible point that achieves the desired outcome for a given point. Suppose the data-points come from $\mathbb{R}^d$, then one possible encoding of the input observation is concatenating the two points into a $2d$ vector.

- Decision Sets: Suppose we are trying to encode an interpretable decision set [28], where each rule is written in if-else form and the data-points come from $\mathbb{R}^d$. One possible encoding of each rule is via a $2d$ vector where each feature has 2 slots (one for the equivalence operator and one for the value) and the slots would be filled in accordingly if the feature appeared in the rule. This way, if the interpretable decision set consists of $k$ rules, then the final encoding would be of the size $k \times 3d$.

## F  Experimental Details

### F.1  Dataset preprocessing

For forward simulation and data bugs, we used the UCI Adult ("adult") dataset as done in [20, 16] (which is linked: here [12]). The data was extracted by Barry Becker from the 1994 Census database. The citizens of the United States consent to having their data included in the Census. According to the U.S. Census Bureau, the United States/Commerce grants users a royalty-free, nonexclusive license to use, copy, and create derivative works of the Software. In the adult dataset, we dropped the `finalweight` feature. For all datasets, we one-hot encoded the categorical features and MinMax scaled all features.

For counterfactual reasoning, we construct a 2D synthetic regression dataset inspired by Kumar et al. [25]. We define the use case label as 1 if increasing a data-point's specific feature $\mathbf{x}_2$ increases the regression model prediction $f(\mathbf{x})$ (holding the other feature $\mathbf{x}_1$ constant), 0 otherwise. The data generation and other use case details are described extensively in Appendix D. In the baseline setting, the agent/human receives no explanation. We evaluate the same local explanations as previous use cases (Section 5). We use a Light-GBM Regressor [21] as the base prediction model class for all explanation settings except GAM, in which the GAM is the base model.

### F.2  Agent architecture

We use the same Deepset architecture [46] for all three use cases (though, note that when the input is of set size 1 in the forward simulation and counterfactual reasoning use cases, Deepsets is effectively a standard feedfoward network). The Deepset architecture is characterized by two parts: (1) the feature extractor $\phi$, which encodes each item in the set, and (2) the standard network $\rho$, which takes in a summed representation of all of the representations from $\phi$. The feature extractor $\phi$ that we use

has a sequence of Dense layers of sizes ($m$, where $m$ is the shape of each observation; 200; 100) respectively, where each layer is followed by an ELU activation with the exception of the last layer. The $\rho$ network is also a sequence of Dense layers of size (100, 30, 30, 10), each followed by an ELU activation with the exception of the last layer which is followed by a Sigmoid activation.

### F.3 Missing Values Adjustment

Figure 5 and Figure 6 show our attempt to recreate the original bug [20] as closely as possible. The authors state that they apply the bug to 10% of data-points. However, we observe that the 30% bug for both explanation methods much more closely resembles the figure that was presented in the original paper.

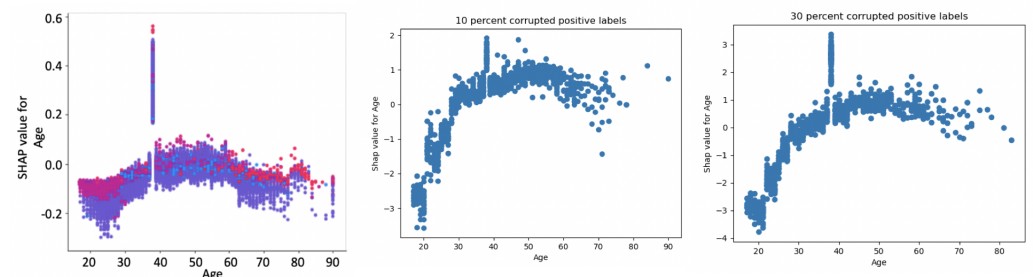

Figure 5: (Left) Image taken from the user interface from Kaur et al. [20] that plots the distribution of SHAP values for the age feature (Middle/Right) Distribution of SHAP values for 1000 points for 10% and 30% corruption.

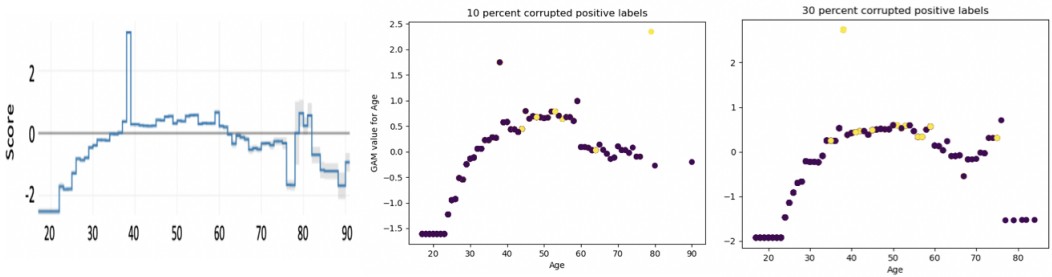

Figure 6: (Left) Image taken from the user interface from Kaur et al. [20] that plots the distribution of GAM values for the age feature (Middle/Right) Distribution of GAM values for 1000 points for 10% and 30% corruption.

### F.4 Base Models

Following Kaur et al. [20], we evaluate both glass-box models that are designed to be interpretable (e.g., GAMs) and post-hoc explanation methods for black-box models (e.g., LIME, SHAP, Anchors). Since GAM is glass-box, the GAM from which local explanations are derived *is* the base predictive model. On the other hand, LIME, SHAP, and Anchors are model-agnostic post-hoc explanation techniques, so they can be used to explain any predictive base model class that is compatible with their Python package implementations. This means that we need to additionally specify the base model family for post-hoc explanation techniques.

Ideally, we would have used the same underlying base predictive model for all explanation settings. When the underlying base model is controlled and held constant, then differences in the agent's accuracy across explanation settings can be attributed to differences in the explanation methods - the variable we're interested in studying - instead of differences in the underlying base models.

However, we encountered the same issue as Kaur et al. [20]: the InterpretML GAM implementation is not supported by the SHAP Python library; thus we could not generate SHAP explanations for

a GAM predictive model. As a result, we also chose to use a different base predictive model class for the 3 post-hoc explanation settings. We emphasize that the comparisons that we make between explanation settings are particular to the base predictive model classes chosen: e.g. between the InterpretML implementation of a GAM and the SHAP Python package used to explain a LightGBM model.

### F.5 Framework Hyperparameters

For all use cases, we generate an observation dataset of up to size 1000 and evaluate on 250 new observations. In our experiments we train our agent with the Adam optimizer with learning rate $10^{-4}$ for our optimizer (and weight decay $10^{-4}$ for counterfactual reasoning), weight decay of binary cross-entropy for the loss function, and batch size of 16. We train for 350 epochs for forward simulation and data bugs and 1000 epochs for counterfactual reasoning.

### F.6 Computational Resources

In general, training an agent using the hyperparameters specified above does not require a GPU, making `SimEvals` on particularly tabular datasets very accessible. When scaling up the data complexity, for example considering image-based data, it may be necessary to use GPUs to train both the prediction model and the agent model.

## G  Full results

We include a more complete set of experimental results for all three use cases. Average agent validation set accuracies (percentages) and their standard error are reported over 3 train-test splits.

### G.1  Forward Simulation

In Hase and Bansal [16], we noticed that the LIME explanation provided included (a) the LIME approximation model's weights, (b) the approximation model's intercept, (c) the sum of the approximation model's weights, and (d) the approximation model's prediction on the point. We also include results for a setting where the agent is given the LIME approximation model's weights only. While the agent's accuracy in this setting is lower, it still outperforms the baseline setting.

Table 5: Error bars when varying the training observations the agent receives to perform forward simulation.

| | Number of Train Set Observations | | | |
| Explanation | 16 | 32 | 100 | 1000 |
| --- | --- | --- | --- | --- |
| LIME | 94.2% ± 3.3% | 99.8% ± 0.2% | 100.0% ± 0.0% | 100.0% ± 0.0% |
| LIME (Weight only) | 86.3% ± 5.5% | 90.8% ± 1.7% | 91.8% ± 1.2% | 96.3% ± 0.2% |
| Anchors | 89.2% ± 2.0% | 93.5% ± 3.4% | 94.7% ± 2.5% | 93.7% ± 1.5% |
| SHAP | 94.5% ± 2.6% | 97.3% ± 0.3% | 99.1% ± 0.6% | 99.3% ± 0.5% |
| GAM | 89.5% ± 9.6% | 96.3% ± 2.3% | 97.4% ± 0.2% | 98.8% ± 0.3% |
| No explanation | 82.3% ± 1.3% | 83.7% ± 1.5% | 85.7% ± 1.8% | 88.7% ± 0.4% |

We note that for forward simulation, there exist known heuristics to predict the model's output using some explanation methods (i.e., it is possible for a human or agent to achieve (near) perfect accuracy). For example, SHAP feature attributions by definition sum to the model prediction [29]. With sufficient examples ($N > 1000$), the predictiveness of SHAP explanations for this use case is demonstrated by the agent's near perfect accuracy.

For completeness we provide the $p$-values from Hase and Bansal [16] which are: LIME (0.014) and Anchors (0.234). Their statistical analysis only compares each explanation to the baseline of no explanation, so the $p$-values provided signify whether the difference in test accuracy of the human when given an explanation versus not is statistically significant.

## G.2 Counterfactual Reasoning

We provide results for the counterfactual reasoning use case for `SimEvals` as well as our MTurk study. We find that `SimEvals` performs well using the LIME explanation which is what we observe in human performance as well.

Table 6: We vary the number of training observations the agent receives to perform counterfactual reasoning.

| Explanation | Number of Train Set Observations | | | | |
|---|---|---|---|---|---|
| | 4 | 16 | 64 | 100 | 1000 |
| LIME | $92.9\% \pm 1.7\%$ | $94.5\% \pm 2.9\%$ | $98.8\% \pm 0.1\%$ | $99.2\% \pm 0.8\%$ | $99.7\% \pm 0.1\%$ |
| SHAP | $56.3\% \pm 2.3\%$ | $52.5\% \pm 1.3\%$ | $55.3\% \pm 1.8\%$ | $57.3\% \pm 2.5\%$ | $64.9\% \pm 3.1\%$ |
| GAM | $56.0\% \pm 2.1\%$ | $53.5\% \pm 1.8\%$ | $56.1\% \pm 2.9\%$ | $57.3\% \pm 3.2\%$ | $63.5\% \pm 1.8\%$ |
| Model Prediction | $52.1\% \pm 1.6\%$ | $55.6\% \pm 1.7\%$ | $54.7\% \pm 2.7\%$ | $57.9\% \pm 2.4\%$ | $60.3\% \pm 2.9\%$ |

Table 7: The average user accuracy on 15 train and test observation along with standard error for counterfactual reasoning use case where for each explanation setting we recruited $N = 20$ Turkers.

| Explanation | Train | Test |
|---|---|---|
| LIME | $66.6 \pm 23.2\%$ | $69.4 \pm 28.2\%$ |
| SHAP | $50.3 \pm 8.9\%$ | $41.4 \pm 14.0\%$ |
| GAM | $48.7 \pm 12.1\%$ | $45.7 \pm 16.4\%$ |
| No Explanation | $58.4 \pm 15.4\%$ | $48.6 \pm 12.8\%$ |

Table 8: Pairwise comparisons using Tukey's HSD between explanation conditions for the counterfactual reasoning MTurk study, showing p-values. We consider $p < 0.05$ to be statistically significant.

| | LIME | SHAP | GAM |
|---|---|---|---|
| LIME | | | |
| SHAP | **0.0017** | | |
| GAM | **0.0016** | 0.906 | |
| No Exp | **0.007** | 0.996 | 0.665 |

## G.3 Missing Values

We provide results for the missing values data bug use case for `SimEvals` as well as our MTurk study. We find that `SimEvals` performs well using the SHAP and GAM explanation which is what we observe in human performance as well.

Table 9: Recreating Kaur et al. [20]'s missing values setting, showing that both SHAP and GAM were successful in finding the bug with high accuracy. We additionally vary the size of the observation set $S$.

| Explanation | Observation Set Size | | | |
|---|---|---|---|---|
| | 1 | 10 | 100 | 1000 |
| SHAP | $63.2\% \pm 2.4\%$ | $84.0\% \pm 1.2\%$ | $99.8\% \pm 0.2\%$ | $100.0\% \pm 0.0\%$ |
| GAM | $64.8\% \pm 3.1\%$ | $87.7\% \pm 3.1\%$ | $100.0\% \pm 0.0\%$ | $100.0\% \pm 0.0\%$ |
| LIME | $55.2\% \pm 1.4\%$ | $56.9\% \pm 1.3\%$ | $64.5\% \pm 1.5\%$ | $75.9\% \pm 0.8\%$ |
| Model Prediction | $57.5\% \pm 1.4\%$ | $57.4\% \pm 1.2\%$ | $58.2\% \pm 2.0\%$ | $67.3\% \pm 11.2\%$ |

Table 10: Standard errors when varying the strength of the missing values bug for a fixed observation set size of 1000.

| Explanation | Bug Strength | | | |
| | 5% | 10% | 20% | 30% |
| --- | --- | --- | --- | --- |
| SHAP | 81.25% ± 3.6% | 82.3% ± 28.0% | 100.0% ± 0.0% | 100.0% ± 0.0% |
| GAM | 58.7% ± 0.8% | 75.2% ± 21.6% | 100.0% ± 0.0% | 100.0% ± 0.0% |
| LIME | 60.9% ± 2.4% | 61.8% ± 0.4% | 60.2% ± 4.7% | 68.7% ± 11.9% |
| Model Prediction | 54.4% ± 2.0% | 56.1% ± 2.6% | 56.6% ± 2.9% | 65.0% ± 12.4% |

Table 11: The average user accuracy on 15 train and test observation along with standard error for data bugs (missing values) use case where for each explanation setting we recruited $N = 20$ Turkers.

| Explanation | Train | Test |
| --- | --- | --- |
| SHAP | 60.7% ± 19.4% | 67.4% ± 27.1% |
| GAM | 58.1% ± 16.2% | 64.4 ± 15.6% |
| LIME | 53.3% ± 11.2% | 48.0% ± 12.3% |
| Model Prediction | 46.7% ± 13.7% | 40.7% ± 11.9% |

Table 12: Pairwise comparisons between explanation conditions for the missing values (data bugs) MTurk study, showing p-values. We consider $p < 0.05$ to be statistically significant.

| | LIME | Model Prediction | SHAP |
| --- | --- | --- | --- |
| LIME | | | |
| Model Prediction | 0.615 | | |
| SHAP | **0.041** | **0.001** | |
| GAM | **0.036** | **0.001** | 0.998 |

## G.4 Redundant Features

To create a setting where there is no bug, we randomize the value of one of the features so there is no correlation between the two and only one feature contains the original information. Note that, in the original user study, the authors do not vary the *strength* of the bug, but we are easily able to study this variant of the bug with our algorithmic framework. We increase the strength of the bug by increasing the number of data-points in dataset $\mathcal{D}_i$ that have the two correlated features.

The agent's accuracy was near random guessing when given LIME explanations or the model prediction baseline, suggesting that neither would be helpful for a human user.

Table 13: We vary the strength of the redundant features bug on the Adult dataset for a fixed observation set size of 1000 and corroborate results from Kaur et al. [20].

| Explanation | Bug Strength | | | | |
| | 10% | 30% | 50% | 70% | 90% |
| --- | --- | --- | --- | --- | --- |
| SHAP | 74.7% ± 2.5% | 78.3% ± 2.6% | 87.5% ± 3.7% | 96.8% ± 2.5% | 99.7% ± 0.2% |
| GAM | 56.5% ± 1.2% | 58.9% ± 2.3% | 84.0% ± 26.9% | 69.8% ± 25.7% | 71.9% ± 24.7% |
| LIME | 56.7% ± 1.3% | 57.3% ± 2.5% | 54.5% ± 2.6% | 54.8% ± 3.9% | 55.3% ± 1.3% |
| Model Prediction | 57.6% ± 3.0% | 56.6% ± 2.0% | 53.7% ± 2.9% | 55.2% ± 0.8% | 58.8% ± 2.7% |

## G.5 Ablation results

**Single data-point observation:** Counterfactual reasoning is one of the use cases where the agent is presented with a single data-point (and explanation). In the main text, we presented results using the

DeepSet architecture. Here, we swap out DeepSet with a non-neural network architecture, LightGBM. We find that the rankings of the explanations are consistent (e.g., compare Table 6 and Table 14).

Table 14: Counterfactual reasoning ablation: we switch out the DeepSet architecture for LightGBM. We find that the rankings are still consistent with the DeepSet results.

| | Number of Train Set Observations | | | | |
|---|---|---|---|---|---|
| Explanation | 4 | 16 | 64 | 100 | 1000 |
| SHAP | 48% | 48% | 47% | 52% | 58% |
| GAM | 48% | 48% | 49% | 50% | 57% |
| LIME | 48% | 48% | 99.6% | 99.6% | 99% |
| Model Prediction | 48% | 48% | 48% | 53% | 52% |

**Multiple data-point observation.** Data bugs is a use case where the user receives multiple data-points as the observation. We do not include these results, but we find low accuracy when using non-neural network models like LightGBM. However, we find similar results when ablating the neural network architecture. For example, instead of summing the representations (as is done in DeepSet), we instead concatenate representations. We find comparable results (with the caveat that one might need to scale up the number of training points in the dataset). For example, for SHAP with set size 100, one would need 4.5x larger dataset to achieve the same accuracy as the original Deepset model (99.8%): 1x data (57.9%), 1.5x data (78.9%), 2.5x data (92.3%), 3.5x data (95.6%), 4.5x data (99.5%). This is because concatenating representations drastically increases the model parameter size, particularly for larger set sizes. We also find that the agent architecture is not drastically affected by adding/removing a layer (Table 9 vs Table 3)

Table 15: Adding a layer to DeepSet, for bug strength of 30%

| | Observation Set Size | | |
|---|---|---|---|
| Explanation | 10 | 100 | 1000 |
| SHAP | 85.8% | 99.5% | 100% |
| GAM | 90.0% | 100% | 100% |
| LIME | 53.3% | 65.8% | 73.5% |

# H  MTurk Details

## H.1  Participants.

A total of 80 participants were recruited to complete the task via Amazon MTurk in 4 batches (20 for each explanation setting). We controlled the quality of the Turkers by limiting participation in our MTurk HIT adults located in an English-speaking countries (specifically the United States and Canada) with greater than a 97% HIT approval rate for quality control. Each worker was only allowed to complete the study 1 time. All participants were retained for final analysis. The estimated time to complete the study was 15 minutes. The study took 21 minutes on average (however, this includes any pauses/breaks the Turker might have taken between the two portions of the task). Each worker was compensated 2.50 USD for an estimated hourly wage of 10 USD.

Our study was approved by the IRB (STUDY2021_00000286) for exempt Category 3 research. We did not anticipate any potential participant risks as we described in the IRB: "The risks and discomfort associated with participating in this study are no greater than those ordinarily encountered in daily life or operation of a smartphone or laptop, such as boredom or fatigue due to the length of the questionnaires or discomfort with the software."

## H.2  Study Context.

**Counterfactual Reasoning.** We introduce the Turkers to the use case using the context of furniture pricing: Turkers are shown data about a furniture item (height and length measurements $\mathbf{x}_i$ and the

item's price $f(\mathbf{x}_i)$) and are asked to perform counterfactual reasoning by choosing whether or not increasing the item's length would increase its price (a screenshot of the interface is shown in Figure 8). The Turkers are instructed to set aside any prior knowledge they may have about furniture pricing and only use the information provided during the study.

**Data Bugs.** We introduce the Turkers to the use case using the context of identifying "bugs" or problems in an Artificial Intelligence system which is used to make predictions about individual incomes. Turkers are shown scatter plots that follow the presentation from the original user study conducted by Kaur et al. [20], reflecting how the system is making predictions along each attribute. The user is provided a demo showing that, for example, as the amount of education a person receives increases, the system deems that attribute to be more important to income prediction. The participants are then asked to determine whether the system contains a "bugs" and learn from the feedback. A screenshot of the interface is shown in Figure 10.

### H.3   Study Interface.

Participants were first presented with brief information about the study and an informed consent form. To complete the HIT, participants were instructed to finish both parts (corresponding to a "Train" and "Test" phase) of the HIT via external URLs to Qualtrics surveys. Both phases of the MTurk evaluation task described in the main text were conducted entirely on Qualtrics.

In both phases of the study, participants are presented with a series of furniture items, and are instructed to answer survey questions where they predict if increasing the item's length will increase its price. Figure 8 illustrates how the information shown to the user varies across different explanation settings.

After opening the Train survey, participants are presented with instructions about the counterfactual reasoning task (Figure 7). After reading the instructions, participants begin the Train phase of the study, where they receive feedback after submitting their responses (Figure 9). Participants then complete the Test phase of the study, where they are not given any feedback after submitting their responses (but are presented with an otherwise identical interface to the Train phase).

For each question, we recorded the participants' response and the time the participant took to respond. At the end of the Test phase, we asked participants to describe the strategy that they used to answer the questions. We also included an attention question where we asked the Turkers to list the two measurements provided. We found that all Turkers either fully answered this question correctly (by saying Height and Length) or gave a response that was reasonable (by saying Price and Length).

In this HIT, we will give you information about furniture items. Each furniture item will have some information:

- the item's **measurements** (Height, Length)
- the **value** of each measurement (this item's height is **112**)
- an **importance** score for each measurement (height has an importance of **-8.3** for the price in this example).
- the **price** for the item (this item's price is **$28**)

| Measurement | Value | Imp |
|---|---|---|
| **Height** | 112 | -8.3 |
| **Length** | 118 | 6.3 |
| **Price** | $28 | |

In part 1 of this HIT, your goal is to **learn to predict** whether **increasing the Length of a given furniture item will increase its price**. The correct answer will differ between furniture items. You should use the information provided to learn to predict the correct answer over time.

Note: The importance score (Imp) represents the contribution that the measurement has on that furniture item's price. The importance scores are different for each furniture item.

This HIT is designed so that **you should not use any prior knowledge you may have about furniture pricing**. After every few questions, you will get feedback on whether your predictions are correct. There will be 15 total practice questions.

In part 2, you will use your practice to make predictions about new furniture items. Your goal is to maximize your number of correct predictions. There will be 15 total test questions.

→

Figure 7: Instructions given to participants in the setting where **explanations are provided** upon opening the Qualtrics survey for the Train phase. The instructions (a) introduce the furniture pricing task and terminology used in the study, (b) show and explain how to interpret an example observation, (c) instruct the Turkers to complete the task by learning from feedback given (and to not use their prior knowledge), and (d) describe the two-phase format of the study. Note that the instructions given to participants in the **baseline setting** where explanations are not provided is the same as the above except that the example observation *does not* have an importance score column and the term "importance score" is also never defined.

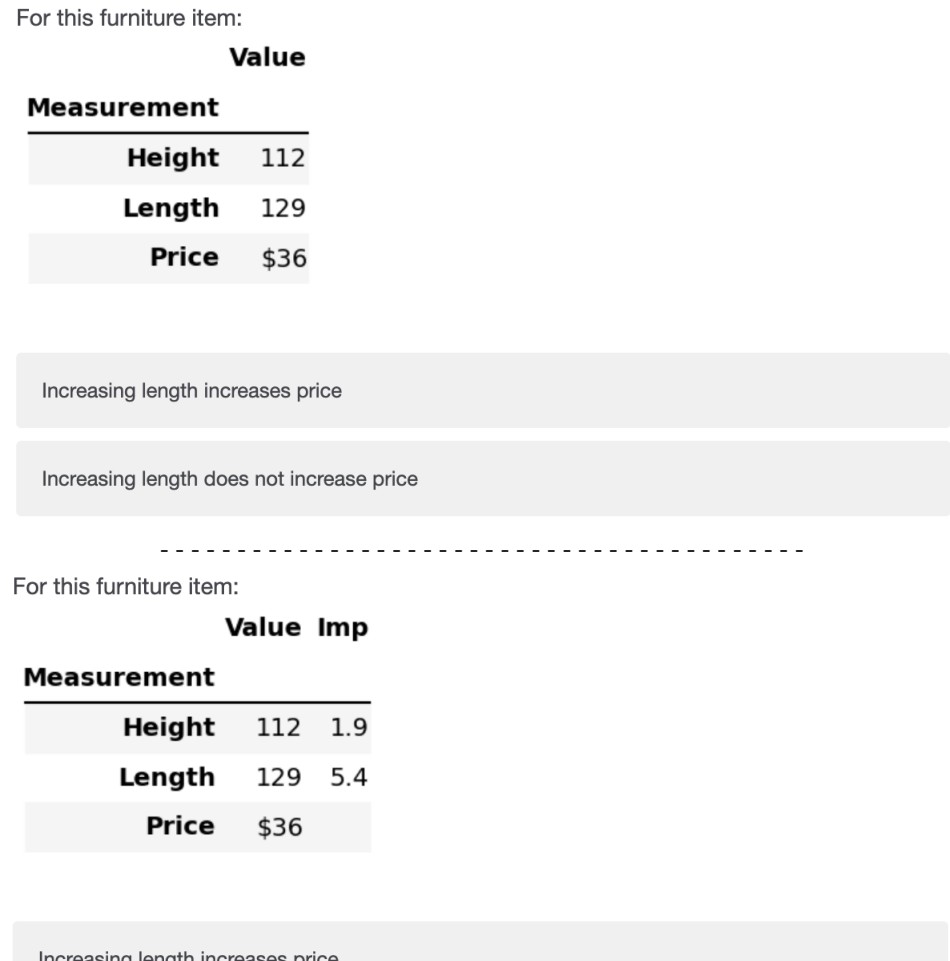

Figure 8: The information provided to participants varies for the Baseline, No Explanation (Top) and the LIME (Bottom) explanation settings. For the LIME, SHAP, and GAM explanation settings, the participant is given the explanation's assigned importance score for each feature in the "Imp" column. Participants are not provided with feature importance scores in the Baseline, No Explanation setting. Each participant is assigned to 1 explanation setting for the entirety of the study, and so will only ever see 1 of these possible interfaces.

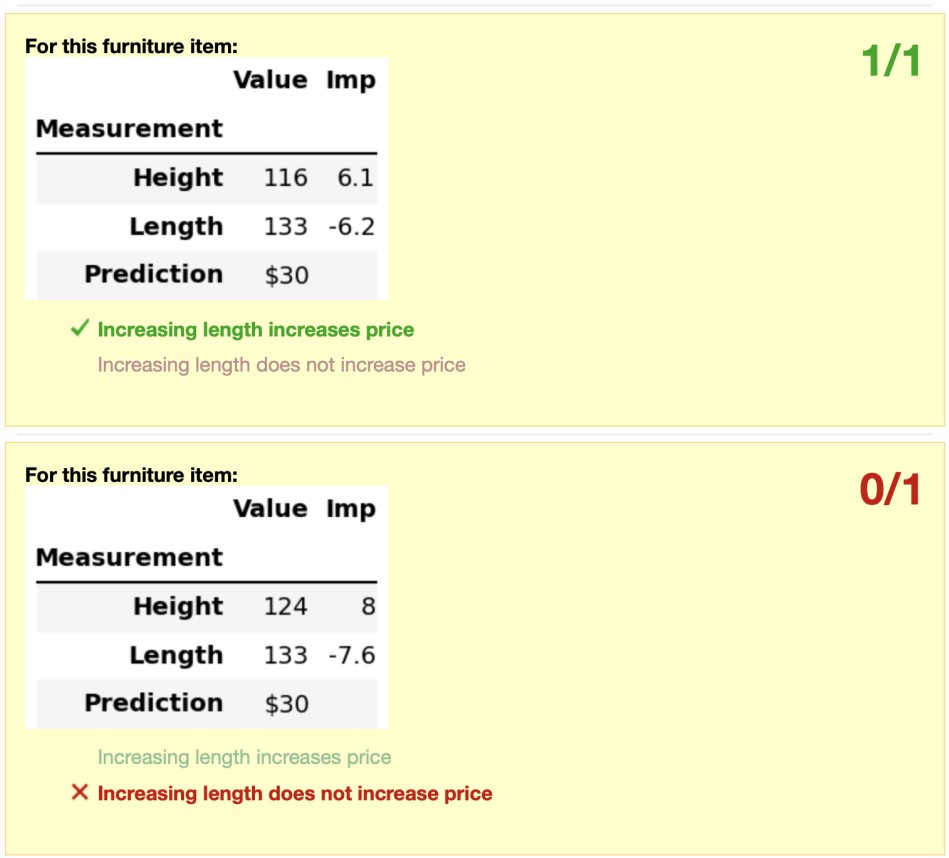

Figure 9: In the Train phase only, participants are provided with feedback on their responses for each furniture item. After submitting their responses, participants learn if their response is correct (Top Example) or incorrect (Bottom Example). Participants may only submit 1 response per furniture item (e.g. they cannot modify their response after it has been submitted), and receive feedback after submitting responses for all 5 observations on each page.

In this HIT, you will be reasoning about different Artificial Intelligence (AI) systems which make predictions about **income** using multiple *attributes*: Age, Work class, Education, Education number, Marital Status, Occupation, Relationship, Race, Sex, Capital Gain, Capital Loss.

For each question, we will give you a set of plots generated by some AI system. An individual plot will show the contribution that a particular attribute has on the AI system's prediction if one were to increase or decrease the attribute.

For example, in this plot, as **education num** *increases*, the contribution that this attribute has on the AI system's prediction *increases*.

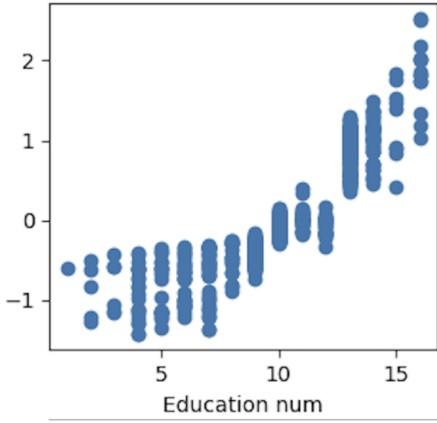

In part 1 of this HIT, your goal is to **learn to predict** whether **there is a "bug"/problem in the given AI system using the provided plots**.

Each question corresponds to a *different* AI system, so the correct answer will differ between questions. You should use the provided plots to learn to predict the correct answer over time. After every few questions, you will get feedback on whether your predictions are correct. There will be 15 total practice questions.

In part 2, you will use your practice to make predictions about new sets of plots. Your goal is to maximize your number of correct predictions. There will be 15 total test questions.

Figure 10: For the data bugs user study, we followed a similar introduction to the task as in the counterfactual reasoning set-up, but instead of a furniture task we tell the Turkers they are finding problems/bugs in an Artificial Intelligence system.

Does the AI System have a bug?

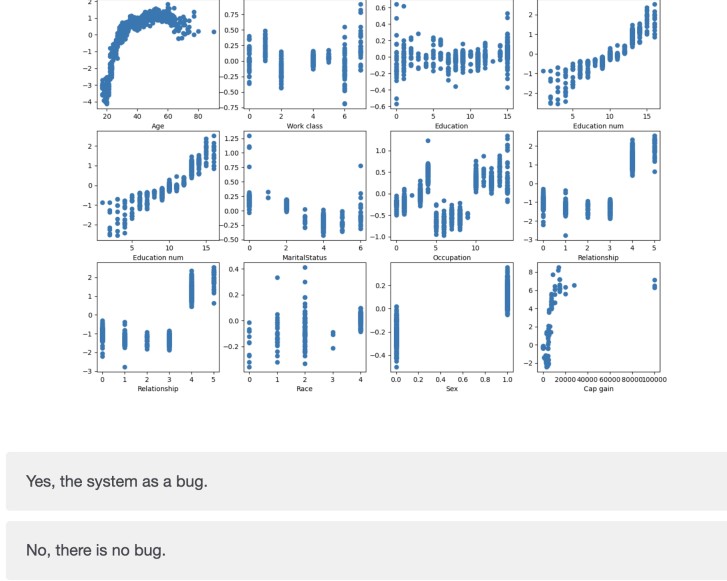

Yes, the system as a bug.

No, there is no bug.

Figure 11: Following the visualizations presented in Kaur et al. [20], we aggregated the explanations of set size $S = 1000$ along each attribute. Specifically, we subset the explanation score for each attribute and plot it against the attribute's values.

### H.4 Pilot Studies

We conducted several pilot human subject studies that informed the design of our final MTurk study. Our pilot studies showed that subjects struggled to complete the counterfactual reasoning task when presented with data $\mathbf{x}$ with a 3 or more features, likely due to information overload [35]. Thus we chose to construct a 2D dataset for our final study.

Our pilot studies also showed that instructions presented to subjects influenced the strategy that they used to respond to the questions. Particularly, we noticed that the way we defined *importance scores* influenced how participants used the scores when answering each question. In an initial trial, we provided participants with the following more detailed definition of importance scores:

- `A positive importance score means that the measurement's value`
  `contributes to the furniture item having a high price.`

- `A negative importance score means that the measurement's value`
  `contributes to the furniture item having a low price.`

- `An importance score with a large magnitude (absolute value)`
  `means that the measurement's value has a larger contribution to`
  `determining the item's price.`

However when we reviewed the participants' self-reported descriptions of the strategy they used, several participants reported that they "used the strategy provided by the instructions" rather than learn from the feedback given during the Train phase. While the given instructions do not directly state a strategy for how the importance scores should be used for the task when literally interpreted, the instructions can be *misinterpreted* to mean that a positive length importance score implies that the item's price will *increase* when its length is increased. Indeed, this is the strategy that many participants used when given the above instructions. As such, we decided to provide a more minimal description of importance scores in the final study (shown in Figure 7) to encourage participants to

learn their own strategy using the information provided as opposed to simply following the given instructions.

# I  Image Experiment

We also conducted `SimEvals` for a user study which evaluated saliency maps for model debugging [2]. We provide this example primarily to illustrate how to perform `SimEvals` on saliency maps and chose not to include these experiments in the main text because there were no positive results for saliency maps (e.g., providing the model prediction alone was enough to detect the bug). The base dataset for this user study was a dog breed dataset (the image label is the breed of the dog). The authors conducted a user study on 4 different bugs: out-of-distribution input, label errors, spurious correlations, and top layer randomization. The authors find that for all of the use cases except for spurious correlations, users relied on *the model prediction* to detect the bug as opposed to the *explanation*. Unfortunately, due to the lack of available code, we were unable to recreate the same spurious correlations that were studied in the paper.

We trained `SimEvals` on the label error bug. We followed a similar data generation process as in the Data Bugs use cases described in the main text and introduced label errors by randomizing the labels before the prediction model was trained and attempt to recreate the prediction model described in the paper. We instantiated the agent architecture using a convolutional Deepset architecture by modifying the $\phi$ network to be a series of convolutions.

The architecture for the agent that is provided with saliency maps: For both the saliency map and the original image, we pass each through a ResNet50 model as a feature extractor, flatten, dropout with probability 0.5, Linear layer with 128 nodes and ReLU non-linearity. We concatenate the two vectors and then pass through another Linear layer with 100 nodes and ReLU non-linearity, and finally a Linear layer with 1 output and Sigmoid non-linearity.

The architecture for the agent that is provided with only the model prediction: For the original image, we pass each through a ResNet50 model as a feature extractor, flatten, dropout with probability 0.5, Linear layer with 128 nodes and ReLU non-linearity. The model prediction is encoded as a one-hot vector and is passed through a Linear layer with 100 nodes and ReLU non-linearity. We concatenate the two vectors and then pass through another Linear layer with 100 nodes and ReLU non-linearity, and finally a Linear layer with 1 output and Sigmoid non-linearity.

We use batch size of 16, Adam optimizer with learning rate 0.001, binary cross entropy loss, and train for 10 epochs.

*Findings:* We find that the agent is able to achieve $> 95\%$ accuracy when provided with the model prediction. This makes sense because provided that the model has a reasonable mental model of classifying which breed a dog is, then it is easy to tell whether the image is correctly classified or not. However, when provided with saliency maps (e.g., Gradient, Integrated Gradient), the agent accuracy is $50 - 65\%$, despite attempts to ablate the CNN architecture. After visual inspection (similar to Figure 13 of their paper) of the saliency map for both cases, it is not particularly evident why it would be helpful in distinguishing when there is a label error.

# J  Discussion: The Agent-Human Gap

## J.1  Human Factors

We describe several human factors that should be considered when researchers are designing `SimEvals` and interpreting a `SimEval` agent's accuracy. We also discuss how we considered these factors when designing our own MTurk study.

**Proxy Metrics.** Since `SimEvals` is agnostic to whether a provided explanation is faithful or not to the model that it is explaining. A researcher using `SimEvals` to select candidate explanation methods should be careful to check that the explanations under consideration also satisfy desirable proxy metrics. This way one would not select explanations that may inadvertently mislead the users. Important proxy metrics include faithfulness to the model [37, 34] and stability over runs [4, 3, 15]. In our user study, we computed explanations using well known, open-source explanation methods

and ensured not to "adversarially" select explanations which would mislead the humans but enable the agent to achieve high accuracy.

**Complexity of Explanation.** A well-studied human factor is that more complex explanations will negatively impact a human's ability to use the explanation [35, 26]. Provided with enough data, the algorithmic agent may in some cases perform better with more complex explanations (if the additional dimensions of complexity also contain more predictive information). When designing `SimEvals` and interpreting agent accuracy, one should factor in the complexity of the explanations that are being evaluated. If the explanation is complex and high-dimensional, then one might discount the agent's accuracy as a measure of informing human performance. In our user study, we limited the dataset to 2 dimensions to control explanation complexity and reduce the likelihood that the user experiences cognitive overload.

**User Interfaces.** The way information is presented to the user through the user study interface or visualization of the explanations can also affect a human's ability to make decisions [39]. In our user study, we control these factors by designing a simple MTurk interface using Qualtrics (a widely-used survey platform).

## J.2 Future Work

It is evident that ML models and humans learn and reason differently. While we found overall that the accuracy of an algorithmic agent can be interpreted as a measure of an explanation's utility for humans, we observe in our MTurk Study that there is a gap. Our proposed framework does not intend to measure potential cognitive factors that may affect the ability of humans to use explanations. An interesting and challenging direction for future work would attempt to (a) understand what such factors are, and (b) measure the extent to which a factor is present in the set of candidate explanations. A better understanding of such factors could also inform development of new explanation methods that meet these desiderata.

It's also worth noting that there are many domains/types of data where humans are still much better at learning than machines. In these settings, an algorithmic agent would also fail to measure the utility of the explanation to a human. Future work may include bringing a human in the loop to define human strategies or behaviors that the agent should follow as to make the agent's reasoning more similar to a human's.