# OpenReview forum: "Use-Case-Grounded Simulations for Explanation Evaluation"
_NeurIPS.cc/2022/Conference — NeurIPS 2022 Accept_

### Official Review · Reviewer_hdF6 · 2022-06-22

**Rating:** 6
**Confidence:** 3
**Soundness:** 3 good
**Presentation:** 3 good
**Contribution:** 3 good

**Summary:**

The paper presents a framework called "SimEvals" for conducting use-case grounded algorithmic evaluations of information content presented in a user study. SimEvals involves training an agent that learns to predict the ground truth label given that the same information would be presented to a human user. The test accuracy of the trained agent is used as a measure for the effectiveness of the information content provided.

**Questions:**

What were the demographics of the AMT workers who participated in the user study? Would this not have a bearing on the effectiveness of the validation and in turn the proposed framework?

How was it ensured that presenting the AMT workers with the set of scenarios similar to the agent's training and validation phase did not induce a "priming effect"?  If the users were not presented with similar situations, is it possible that the results could have changed?

How does the performance of the framework change with the number of explanations--typically even human users experience fatigue/cognitive overload when presented with a number of explanations. So in the light of this, what kind of advantage or performance benefit the proposed framework can provide?





**Limitations:**

Please refer to comments under the last two questions

**Strengths And Weaknesses:**

Strengths:

1. The paper presents an interesting framework that could potentially aid future user studies in terms of assessing which aspects need actual human user evaluation thereby saving manual efforts and costs

2. The framework also expands the scope of XAI studies in the sense those explanation methods which are often overlooked may also find a place through this scheme

3. The method will offer promising pathways to further research in this area

Weakness:

1. Test accuracy of the agent may not be a wholistic measure of the performance, might need to augment with other metrics as well.

2. demographics of the AMT workers could have a bearing on the performance validation scheme

3. Might have to consider the influence of cognitive human biases including aspects related to subjectivity, confirmation bias, priming effects among others in evaluation

4. As the study involved human users, information related to institutional ethical clearance should be included in the main paper.

---

> ### Author Response · Authors · 2022-08-02
> **Response to Reviewer hdF6**
>
> Thank you for your review and for acknowledging the strengths of this work.
>
> We first respond to the weaknesses below:
>
> **Snippet:** *“Test accuracy of the agent may not be a wholistic measure of the performance, might need to augment with other metrics as well.”*
> - We agree that including additional metrics in future work may paint a more complete picture of the algorithmic agent’s behavior. However, we show in our experiments that __test accuracy is an informative measure for selecting explanations__ for a target use case.
>
> **Snippet:** *“demographics of the AMT workers could have a bearing on the performance validation scheme”*
> - We emphasize that our validation scheme also considers findings from the users of the other two studies that we cite: the forward simulation user study (Hase and Bansal) was conducted with undergraduates and undergraduate college students and the data bugs user study (Kaur et al) was conducted with data scientists.
> - Additionally, while there may be variability between individual AMT workers’ demographics that may affect their ability to complete the task, we account for this variability by assigning Turkers to each condition uniformly at random and enforcing that each individual worker can only complete the HIT once.
>
> **Snippet:** *“Might have to consider the influence of cognitive human biases including aspects related to subjectivity, confirmation bias, priming effects among others in evaluation”*
> - Please see __Point #3__ in the general response.
>
> **Snippet:** *“information related to institutional ethical clearance should be included in the main paper.”*
> - We will move information about the IRB to the main text; it is currently in Appendix H.1.
>
> We now respond to the questions:
>
> **Snippet:** *“What were the demographics of the AMT workers who participated in the user study? Would this not have a bearing on the effectiveness of the validation and in turn the proposed framework?”*
> - We recruited AMT workers by filtering for those with a higher than 97% acceptance rate and those from US or Canada, and otherwise used all default settings without applying additional selection criteria. We responded to the second question above.
>
> **Snippet:** *“How was it ensured that presenting the AMT workers with the set of scenarios similar to the agent's training and validation phase did not induce a "priming effect"? If the users were not presented with similar situations, is it possible that the results could have changed?”*
> - We ensured that each experimental condition used the __same__ instructions, interface, and formulation to avoid “priming” users. Thus, the differences we observe in user performance should be attributed to the usefulness of the information contained in the explanation itself.
> - We intentionally have the same “scenario” in training and validation for the AMT workers intentionally so that the user, like the algorithmic agent, can learn some heuristic in train and apply it in validation. This is what user studies like Hase and Bansal do.  It could be interesting in future work (whether through simulations or user studies) to look at how well agents/humans can generalize to even more dissimilar “scenarios”.
>
> **Snippet:** *“How does the performance of the framework change with the number of explanations…what kind of advantage or performance benefit the proposed framework can provide?”*
> - While the performance of the algorithmic agent may not change in the same way as the performance of a human user as we increase the number of explanations (e.g., the training set size), we clearly state in L56-57 that the goal of SimEval is not to attempt to model such human factors which may include fatigue or cognitive overload. We explicitly encourage researchers to consider these additional human factors when interpreting SimEval results (see L153-158 of initial submission).
> - We believe that we demonstrate SimEvals can still be very helpful for user study design, as explanations that are not predictive for a use case will not be helpful to humans, regardless of these additional human factors.

---

### Official Review · Reviewer_jL67 · 2022-07-11

**Rating:** 6
**Confidence:** 4
**Soundness:** 3 good
**Presentation:** 4 excellent
**Contribution:** 2 fair

**Summary:**

In this paper the authors propose a framework for simulated evaluations (SimEvals) meant to help guide explainable AI (XAI) researchers in terms of what explanations to include in a human subject study. The authors introduce the framework, which involves preparing a dataset with the outputs of the base model (the model that we want to explain) along with any explanations and then training another model as a soft surrogate for human evaluation. The authors go on to show how the performance of the second model on a test set can be seen to roughly correspond to human performance in two existing datasets and then in the results of one novel human subject study.


**Questions:**

1. Why not run a correlation test between the human predictions and the agent’s predictions on the same problems?
2. What use cases that weren’t already possible or what benefits does the model confer?


**Limitations:**

The authors overall do a good job expressing the limitations of their approach (particularly in the appendices). However, one issue they do not discuss is the possibility of a SimEvals-like approach being used to avoid user studies entirely. While the authors argue against this (“We note that the agent’s performance (i.e. test set accuracy) with a given explanation method is not intended to be interpreted as an anticipated human subject’s performance.”), other claims in the paper would seem to support this use case (“In our experiments, we show that the relative ordering of explanation method performance is consistent for SimEvals and human subject evaluations.”). Some discussion about how this could be avoided in the XAI research field would be appreciated.

**Strengths And Weaknesses:**

The primary strengths of the paper are its formalization of the SimEvals framework, including the guidance and discussion around how and when to apply the framework, and the results section. While other researchers have run experiments that can be understood as instantiations of this framework, to the best of this reviewer’s knowledge, there have been no existing attempts to define this particular framework. That is therefore novel. The results do also demonstrate some value to this approach, though I do have concerns with them that I’ll address below.

I have two major concerns in terms of the weaknesses of this paper, and they relate to the strengths above. First, there’s the framework. While this is novel experiments like those the framework suggests have been run in the past. As such, the question then turns to the strength of this formalization. In other words, how much is this formalization likely to help other researchers relative to just continuing to have researchers run one-off, domain-dependent experiments? This is unclear to me after reading the paper, particularly because much of the framework is itself domain-dependent. The authors speak to this point in the related work and Appendix A but there’s still no specific argument as to the benefits of this framework in comparison to running domain-dependent versions of these experiments without the framework. The closest we get is that the authors hope that this framework/paper will help future researchers run these experiments, which is certainly possible. But I’m not sure if this means that future researchers will run these experiments faster, be more likely to run this kind of experiment, or so on.

My second major concern is with the results, which appear to be somewhat weak. At best the results seem to match in terms of the rough ranking (particularly the first place) when comparing user accuracy to agent accuracy. However, there’s cases where the ranking doesn’t match up. For example in Table 3, where the agent is able to get useful information from SHAP and GAM, while these explanations appear actively harmful in the average human case. These results also do not support the claims made by the authors (e.g., “we find strong correspondences between the SimEval agents’ and humans’ accuracy on the use case”). Given the claims, I was expecting something more like a correlation test (e.g., Spearman’s rho) comparing the individual human and agent predictions for the specific individual cases. If there’s some reason such a test would be inappropriate, it would be helpful for the authors to offer an explanation. But the average accuracy doesn’t seem to be particularly representative of human performance, as the authors themselves indicate given the high-variance of LIME results in Table 3.

Overall, I’m torn between believing there is value to this paper (and more value in the underlying work), but having that tempered by the weak results that do not match up with the claims, and the lack of clarity around what use cases the framework supports over the existing examples. I’ve settled on a weak accept for now but it’s a shaky weak accept.

---

> ### Author Response · Authors · 2022-08-02
> **Response to Reviewer jL67**
>
> Thank you for your review. We respond to your two concerns below, which we believe also address your questions:
>
> **Snippet:**  *“While this is novel, experiments like those the framework suggests have been run in the past… How much is this formalization likely to help other researchers relative to just continuing to have researchers run one-off, domain-dependent experiments?”*
> - Please see __Point #1__ in the general response for how our formalization is concretely different from prior algorithmic evaluations along multiple dimensions.
>
> **Snippet:**  *"But I’m not sure if this means that future researchers will run these experiments faster, be more likely to run this kind of experiment, or so on.”*
> - We believe this work will help researchers be more likely to run algorithmic evaluations. Presently, researchers who run human subject studies to evaluate explanations (e.g. [14, 16, 18, 20, 35] from our citations) by and large do not run any algorithmic evaluation(s) to inform their user study.  We hypothesize this is because of the lack of a framework that is generalizable across use cases and explanations and of guidance on how to run such evaluations, which is exactly the problem our work addresses.  We are also working on a tutorial as a resource for researchers.
>
>
> **Snippet:**  *“At best the results seem to match in terms of the rough ranking (particularly the first place) when comparing user accuracy to agent accuracy. However, there’s cases where the ranking doesn’t match up”. *
> - Thank you for bringing up this concern, and please see __Point #2__ in the general response where we clarify our contribution claim and demonstrate how our results strongly support that claim. Thus, we believe spearman’s rank correlation would not be an appropriate test (furthermore, such a test would not work well for small size N, where N = 3 or 4 for our use cases).
>
> **Snippet:**  *“The possibility of a SimEvals-like approach being used to avoid user studies entirely.”*
> - Please see our response to __Point #3__ in the general response.

---

### Official Review · Reviewer_Diw4 · 2022-07-11

**Rating:** 8
**Confidence:** 3
**Soundness:** 4 excellent
**Presentation:** 4 excellent
**Contribution:** 4 excellent

**Summary:**

The authors propose a procedure for evaluating explanations using "simulated evaluations" (SimEvals). The authors demonstrate SimEval using common explanation methods, and demonstrate that their findings align with (human) user responses from a small MTurk survey.

**Questions:**

See above

**Limitations:**

Yes

**Strengths And Weaknesses:**


This paper clearly presents an idea that I believe is an important contribution to the XAI literature. Their simluations and user study are both relatively small, but their results are convincing. It should be noted that many other researchers have thought about using downstream/user behavior to guide the design of "good" explanations. To my knowledge these studies live mainly in the HCI space. I think it's important for the authors to engage with this literature, because it is very similar in motivation to this work. Some examples:

- https://doi.org/10.1109/TREX51495.2020.00005
- https://doi.org/10.1145/3301275.3302265

It's interesting that the authors chose to frame their method as an alternative or supplement to user studies. I could imagine a version of SimEval that exists completely independently of users, where the purpose of SimEval is to assess how informative an explanation is about a particular model behavior---which is itself a prediction problem. (SimEval does this, but framed in terms of users.)

Since the authors choose to center on users, this raises some important questions:

1) What if the SimEval Agent's behavior differs from the user's behavior? That is, even if an explanation is informative (in a statistical or ML sense) about some aspect of a model, how do we know that this information will also be useful for a human user? I would like to  see the authors engage with this question somewhere in the main body of their paper.

2) The SimEval framework assumes that the evaluator knows what "good" and "bad" downstream behavior looks like. This is not always the case---the three use cases presented here (forward simulation, counterfactual reasoning, and data bugs)are certainly well-motivated, but what about use cases that aren't easily quantified? For example with debugging, we often don't know what the source of a bug is! Is this framework still useful in this case?

---

> ### Author Response · Authors · 2022-08-02
> **Response to Reviewer Diw4**
>
> Thank you for your review. We are excited that the reviewer believes that this work “is an important contribution to the XAI literature”. We respond to your comments below:
>
> **Snippet:** *“I think it's important for the authors to engage with [the cited] literature”.*
> - Thank you for the pointers to these papers (the first of which is an XAI user study and the second of which is a position paper motivating more utility-focused evaluations of explanations). We added them to our related work.
>
> **Snippet:** *“I could imagine a version of SimEval that exists completely independently of users, where the purpose of SimEval is to assess how informative an explanation is about a particular model behavior”.*
> - We agree with the reviewer that an interesting direction for future work is to explore how SimEvals, which measure the predictive information in an explanation, could potentially be used for purposes other than human subject study design. We chose to motivate our work with the goal of aiding human subject studies as we believe it is a very timely problem with high impact, as evidenced by the large number of XAI user studies (e.g. [14, 16, 18, 20, 35] from our citations and more in the HCI space as you mention) that we believe could benefit from improved evaluation workflows.
>
> We now address your questions:
>
> **Snippet:** *“What if the SimEval Agent's behavior differs from the user's behavior? That is, even if an explanation is informative (in a statistical or ML sense) about some aspect of a model, how do we know that this information will also be useful for a human user?”*
> - Please see __Point #3__ of our general response.
>
> **Snippet:** *“The SimEval framework assumes that the evaluator knows what "good" and "bad" downstream behavior looks like.…what about use cases that aren't easily quantified?”*
> - We agree with your concern and note that quantitatively evaluating the utility of explanations in such settings is difficult not only to do algorithmically (using SimEvals) but also more broadly in user studies, as it is also unclear how to measure a human’s success in such settings.  We believe that exploring metrics for “success” in such settings is an important direction for future work that can inform how SimEvals can be applied.

---

### Official Review · Reviewer_hGqz · 2022-07-18

**Rating:** 5
**Confidence:** 4
**Soundness:** 2 fair
**Presentation:** 3 good
**Contribution:** 1 poor

**Summary:**

The paper introduces an automatic evaluation framework, called SimEvals, to assess the usefulness of explanations in 3 use cases (forward simulation, model debugging, and counterfactual reasoning). SimEvals mimics user studies by training agents instead of humans. Thus it is cheaper and intended to use before more costly user study.

=====
After authors' response: My questions have been properly addressed and I am satisfied with the addition of the newly performed user study and the other clarifications being added to the paper. I have increased my score accordingly.

**Questions:**

Maybe the authors could give convincing arguments as to why a comparison with human studies (done by the authors, there's no need of replication of previous results) is not necessary for the 2 scenarios where it has not been performed.

**Limitations:**

The authors do not mention anything about potential negative societal impact.

**Strengths And Weaknesses:**

Strengths
The paper addresses a timely and important topic, that of evaluating the usefulness of explanations in 3 real-world scenarios.
It is well written, easy to read, and clearly states its scope.

Weaknesses
The 2 main weaknesses I see are: (1) the lack of necessary extensive experiments, and (2) the unconvincing results even among the performed experiments. Since the evaluation method proposed is not of much technical novelty, the main strength of this work was supposed to lie in showing with **extensive experiments** that the results of SimEvals correlate with human results. However, there are very few experiments: only 3 or 4 different explanations are considered for each of the 3 scenarios. And more importantly, only 1 scenario has a comparison with human studies (Table 3). The authors justify the lack of comparison with human studies in footnote 3 by saying that "(a) their interfaces are not publicly available and (b) we do not have access to the same set of participants to compare our study results to theirs.". However, both arguments are extremely shallow: for (a), they could have devised a reasonable interface to do this; for (b), there's absolutely no need to have the exact same participants to draw the conclusion of whether their automatic evaluation correlates with human studies. Finally, the authors claim that "there is a match in the relative rankings" in Table 3, however, we see that the ordering of SHAP, GAM and No explanation are different in SimEvals than in Human studies, and that the only ordering that matches is in the best explanations (from LIME). The authors mention that "an ANOVA test and found a statistically significant difference between the Turkers’ accuracy using LIME vs. all other explanation settings" but they do not mention if there's a statistical difference between SHAP, GAM, and No explanation in both settings, which, if it is, would have indicated different rankings of the explanatory methods between SimEvals and Humans. Also, the authors motivate that a better hyperparameter search for the explanatory methods can be done using SimEvals, but they do not perform any experiments and leave this for future work. Overall, I find the experimental setup very unconvincing, for a work in which this would have been its main focus.

Minor:
* Tables don't have bold numbers for the best results
L272 evaluates --> evaluate
L280 implements --> implement
Include results from human studies in Table 1
bold results in Table 1
L993 that,
L725&728 ??
L856 Table ??

---

> ### Author Response · Authors · 2022-08-02
> **Response to Reviewer hGqz**
>
> Thank you for the review. We are glad that you agree that our work addresses a “timely and important topic”.  We first respond to your question and then the 2 main weaknesses.
>
> **Snippet:** *“why a comparison with human studies (done by the authors, there's no need of replication of previous results) is not necessary for the 2 scenarios where it has not been performed”*:
> - Our initial submission __did__ compare the algorithmic agent performance to human performance from prior user studies for forward simulation (L252, 262 of original pdf) and data bugs (L274, 284 of original pdf). That said, we agree that these comparisons can be made more clear in the main text. We __added a “Human Test Accuracy” column__ to Tables 1 and 2 and describe these comparisons in more detail in Appendix G.
> - Even without replicating the user studies for the 2 scenarios, we argue that there is value in comparing the results of SimEval to human results from __independently run, peer-reviewed user studies__, showing that SimEvals identifies promising explanations even across a diverse population of study participants.
> - When considering your question, we realized that there would be value in running an additional experiment: __conducting our own user study for the data bugs use case__.  The original study conducted by Kaur et al led participants through a semi-structured exploration of various explanation tools. Thus, the original study did not individually evaluate each explanation’s usefulness in aiding users to identify bugs and did not consider potential baseline explanations to justify the need for explanations.
> - In our new MTurk experiment for data bugs, we follow a similar evaluation procedure as the counterfactual reasoning use case and introduce 2 new experimental settings to the original study. We find that SimEval can again identify promising explanations from unhelpful explanations: based on agent test accuracy, SimEval would select SHAP and GAM over LIME or a Model Prediction baseline, and we find that humans achieve 20% better test accuracy (on average) with SHAP/GAM than with LIME/Model Prediction. Our complete study results are in Table 3 of the revised draft.
>
> We now respond to 2 main weaknesses in light of these results:
>
> (1): *“the lack of necessary extensive experiments”*
>
> **Snippet:** *“Since the evaluation method proposed is not of much technical novelty, the main strength of this work was supposed to lie in showing with extensive experiments that the results of SimEvals correlate with human results.”*
> - We strongly disagree with the comment, please see __Point #1__ in the general response which outlines multiple ways in which our work is novel.
>
> **Snippet:** *“there are very few experiments: only 3 or 4 different explanations are considered for each of the 3 scenarios.”*
> - We respectfully disagree that our work has “very few” experiments and believe that the reviewer’s summary of the experiments oversimplifies our evaluation, where the goal was to provide guidance on how researchers can instantiate SimEvals for 3 diverse use cases and verify those results.
> - In addition to training a SimEval agent for each of the “3 or 4 different explanations” per use case (as was done in previous work), we also studied the effect of varying multiple parameters (e.g., the strength of bug type, number of training set observations, number of data-points per observations, agent architecture) on agent performance.
>
> **Snippet:** *“the authors motivate that a better hyperparameter search … but they do not perform any experiments and leave this for future work”*
> - As stated in the Introduction (L60-62), while the SimEvals framework can be used for hyper-parameter selection, we focused on rigorously evaluating SimEvals for selecting explanation methods as this has been the exclusive focus of existing literature that runs user studies.
>
> (2): *“the unconvincing results even among the performed experiments”*
>
> **Snippet:** *“the authors claim that "there is a match in the relative rankings" in Table 3, however, we see that the ordering of SHAP, GAM and No explanation are different in SimEvals than in Human studies, and that the only ordering that matches is in the best explanations (from LIME)."*
> - Please see **Point #2** in the general response where we clarify our contribution claim and demonstrate how our results strongly support that claim.
>
> **Snippet:** *“they do not mention if there's a statistical difference between SHAP, GAM, and No explanation in both settings”*
> - There is no statistical difference between the average accuracy when given SHAP, GAM, and No explanation for human subjects. We add p-values to Table 8.
> - There are also significant overlaps in the error bars for Agent performance on SHAP, GAM, and No explanation (see Table 6).
>
> **Snippet:** *“The authors do not mention anything about potential negative societal impact.”*
> - Please see our response to **Point #3** in the general response.

---

### Author Response · Authors · 2022-08-02
**General Response**

We would like to thank all reviewers for their invaluable feedback. We appreciate their efforts to strengthen our work. We respond to related comments in a general post:

__Point #1: Novelty + usefulness of framework__ (Reviewer hGqz and jL67):
- As shown in the table which contextualizes related work that we discuss in Appendix A, no prior work both (1) provides a general algorithmic evaluation framework that reflects the diverse set of user studies that a researcher may consider and (2) verifies their proposed framework using multiple human subject studies.

| |Algo Eval: Agent learns how to use explanation (i.e., no heuristic needed)?|Algo Eval: Framework generalizes to different use cases?| Human Eval: Runs a user study?|Human Eval: Agent results match human?|Human Eval: Number of explanations and baselines evaluated?|
|:-:|:-:|:-:|:-:|:-:|:-:|
|Expo|No|No|Yes|No|1 and 1|
|Influence Functions|No|No|No|-|-|
|Student-Teacher Models|No| No| Yes| No| 2 and 1|
|Anchors|No| No| Yes| Yes| 2 and 0|
|User studies e.g.  [14, 16, 18, 20, 35]|-|-|Yes|-|Avg: 2-4 and 1-2|
|Ours (SimEvals)|Yes|Yes|Yes (we run or compare to multiple studies)| Yes|4 and 2|
- In this work, we also provide extensive technical guidance to researchers (Section 3, Appendix E) to illustrate how to generate data and set up SimEvals for a wide variety of explanation types, use cases, and data types.

__Point #2: Contribution framing__ (Reviewer hGqz and jL67):
- Reviewers hGqz and jL67 state our claim that agent and human rankings “match” is inaccurate. We agree that the current phrasing may be misleading, and we have rewritten our contribution as “SimEvals helps distinguish between which explanations are promising vs. unhelpful to humans” (L73).
- When there is a significant gap between SimEval performance on two explanations, we observe a *similarly significant gap in human performance* when using the same explanations. We find that the difference between the “best” (i.e., most promising) explanation(s) and all others is statistically significant. Even though we observe in Table 2 (formerly 3) that the exact ordering of the 3 “worst” (i.e., unhelpful) explanations differs between the agent and humans, these differences are small and not statistically significant (see Table 8).  This same reasoning can be applied to the other use cases as shown in agent vs human results in Tables 1 and 3 of the revised draft.
- More broadly, we believe that researchers should focus on __statistically significant differences__ between explanation methods when interpreting their SimEvals, and we caution researchers from concluding that smaller differences may generalize to human performance (as highlighted in Section 6).


__Point #3: The gap between agent and human__ (all reviewers):
- While SimEvals can identify promising explanations for downstream user studies, we find in our experiments and analyses that raw agent and human performances are not equal. This is unsurprising given that SimEval intentionally captures the predictiveness of the information content and does not model human factors.
- However, given that SimEvals may be adopted in future user study workflows, we agree with the reviewers’ suggestions that we should more clearly outline potential misinterpretations or misuses of SimEvals. We moved the existing discussion to the main text from Appendix J and extended it in Section 6 of the revised draft.

---

### Meta-Review · Area_Chair_FN8p · 2022-08-27

**Recommendation:** Accept
**Confidence:** Certain

**Metareview:**

The paper proposes simulated evaluations (SimEvals) to guide explainable AI (XAI) researchers about what explanations to include in a user study. All the reviewers agreed that this is a novel contribution to a significant and timely problem. There were common questions around the empirical evaluations that the authors clarified during the feedback phase.

The reviewers have acknowledged the authors' responses and have confirmed that their questions were adequately addressed.

By adding the new table contrasting prior work that the authors included in their feedback, as well as the clarifications from the reviewer discussion, the paper will be substantially stronger.

**Award:**

No

---

### Decision · Program_Chairs · 2022-09-14

Accept